# Spatially mapped single-cell chromatin accessibility

Casey A. Thornton[1], Ryan M. Mulqueen[1], Kristof A. Torkenczy [1], Andrew Nishida[1], Eve G. Lowenstein[1], Andrew J. Fields[1], Frank J. Steemers[2], Wenri Zhang[3], Heather L. McConnell [4], Randy L. Woltjer[5], Anusha Mishra[4,6], Kevin M. Wright [7] & Andrew C. Adey [1,6,8,9 ✉]

High-throughput single-cell epigenomic assays can resolve cell type heterogeneity in complex tissues, however, spatial orientation is lost. Here, we present single-cell combinatorial indexing on Microbiopsies Assigned to Positions for the Assay for Transposase Accessible Chromatin, or sciMAP-ATAC, as a method for highly scalable, spatially resolved, single-cell profiling of chromatin states. sciMAP-ATAC produces data of equivalent quality to non-spatial sci-ATAC and retains the positional information of each cell within a 214 micron cubic region, with up to hundreds of tracked positions in a single experiment. We apply sciMAP-ATAC to assess cortical lamination in the adult mouse primary somatosensory cortex and in the human primary visual cortex, where we produce spatial trajectories and integrate our data with non-spatial single-nucleus RNA and other chromatin accessibility single-cell datasets. Finally, we characterize the spatially progressive nature of cerebral ischemic infarction in the mouse brain using a model of transient middle cerebral artery occlusion.

[1] Molecular and Medical Genetics, Oregon Health & Science University, Portland, OR, USA. [2] Illumina Inc., San Diego, CA, USA. [3] Anesthesiology and Peri-Operative Medicine, Oregon Health & Science University, Portland, OR, USA. [4] Jungers Center for Neurosciences Research, Department of Neurology, Oregon Health & Science University, Portland, OR, USA. [5] Department of Pathology, Oregon Health & Science University, Portland, OR, USA. [6] Knight Cardiovascular Institute, Oregon Health & Science University, Portland, OR, USA. [7] The Vollum Institute, Oregon Health & Science University, Portland, OR, USA. [8] CEDAR, Oregon Health & Science University, Portland, OR, USA. [9] Knight Cancer Institute, Oregon Health & Science University, Portland, OR, USA. ✉email: adey@ohsu.edu

Heterogeneous cell types coordinate in complex networks to generate emergent properties of tissues. These cell types are not evenly dispersed across tissues, creating spatially localized functionality. In many disease states, this becomes more apparent, as the affected organ experiences spatially progressive etiologies. For example, following cerebral ischemic injury, astrocytes, and microglia enter reactive states that are metered by proximity to the site of infarction[1], but this spatial information has, so far, been difficult to assess. Single-cell technologies have advanced cell type and state characterization efforts by enabling the isolation of signals from individual cells within a sample, thus resolving the heterogeneity of complex tissues. Applications of single-cell technologies have identified novel cell types with characteristic -omic signatures in the highly complex tissue of the brain[2,3]. In the cerebral cortex, specifically, cells form an intricate layered hierarchical structure comprised of both neuronal and glial cell types that generate sensory, motor, and associational percepts[4]. Layer-specific gene expression profiles of cortical neurons and astrocytes have been characterized by spatial transcriptomic approaches and immunohistochemical (IHC) staining; however, spatially mapped epigenetic states of cortical cells have yet to be directly assayed, without relying on the data integration[5–7].

To address this challenge, several strategies have emerged to assay transcription either directly in situ or in a regional manner. The former techniques utilize fluorescence in situ hybridization (FISH)[8–10] or in situ RNA sequencing[11,12]. While powerful, FISH methods require the use of a defined probe set and are limited to the identification of DNA and RNA sequences. In contrast, technologies that utilize array-based mRNA barcoding do not require a defined set of genes and operate similarly to single-cell RNA-seq methods[13,14], thus allowing for whole transcriptome profiling. Initial iterations of these platforms capture regional transcription over multiple cells; however, higher resolution variants may facilitate single-cell resolution. Unfortunately, these platforms rely on the relatively easy access to mRNA molecules that can be released from the cytoplasm and hybridized to barcoding probes, making the expansion into nuclear epigenetic properties challenging. With the wealth of epigenetic information that resides in the nucleus and the value it can add to characterizing complex biological systems[15–17], we sought to address this challenge by harnessing the inherent throughput characteristics of single-cell combinatorial indexing assays[18,19].

Here, we present single-cell combinatorial indexing from Microbiopsies with Assigned Positions for the Assay for Transposase Accessible Chromatin (sciMAP-ATAC). sciMAP-ATAC preserves the cellular localization within intact tissues and generates thousands of spatially resolved high-quality single-cell ATAC-seq profiles. As with other "sci-" technologies, sciMAP-ATAC does not require specialized equipment and scales nonlinearly, enabling high-throughput potential. Building upon multiregional sampling strategies[20,21], where several regions are isolated, we reasoned that the sample multiplexing capabilities of combinatorial indexing could be utilized to perform high-throughput sampling at resolutions approaching those of array-based spatial transcriptional profiling, all while retaining true single-cell profiles. Unlike multiregional sampling, we perform high-density microbiopsy sampling, ranging from 100 to 500 µm in diameter, on cryosectioned tissue sections, between 100 and 300 µm in thickness, to produce up to hundreds of spatially mapped punches of tissue, each producing a set of single-cell chromatin accessibility profiles (Fig. 1a). We demonstrate the utility of sciMAP-ATAC by profiling the murine and human cortex, where distinct cell type compositions and chromatin profiles are observed based on the spatial orientation of the punches, and further extend the platform to characterize cerebral ischemic injury in a mouse model system, where cell type compositions and epigenetic states are metered by proximity to the injury site (Supplementary Fig. 1).

## Results

### Single-cell combinatorial indexed ATAC-seq from microbiopsy punches.
Single-cell ATAC-seq requires the isolation and processing of nuclei such that the nuclear scaffold remains intact to facilitate library preparation via transposition in situ; it also requires that the chromatin structure is maintained to produce a chromatin accessibility signal. We and others have explored methods for tissue preservation that are compatible with single-cell ATAC-seq[18,22]; however, we sought to confirm that these strategies are compatible with freezing techniques used for cryosectioning and IHC staining of tissue. We tested our workflow on mouse whole brain samples by processing one hemisphere using flash-freezing methods designed for tissue freezing medium (TFM) embedding and cryosectioning ("Methods"), and processing the paired hemisphere as fresh tissue. Our previously established nonspatially resolved sci-ATAC-seq workflow[22] was performed on both hemispheres, including pooling post-transposition for sorting, PCR amplification, and sequencing. Flash-frozen and fresh nuclei produced nearly identical passing reads per cell at the depth they were sequenced, along with comparable fractions of reads present in a set of aggregate mouse ATAC-seq peaks (FRiS; 0.93 and 0.91 for fresh and frozen, respectively; Supplementary Fig. 2a, b).

We then explored techniques for cryosectioning flash-frozen TFM-embedded tissue at thicknesses compatible with microbiopsy punching. Typically, cryosectioning is used to produce sections for imaging applications, and thicker sectioning results in tissue fracture. Drawing on past literature[23], we carried out a series of experiments testing several sectioning thicknesses and punch diameters followed by nuclei isolation and debris-cleanup on flash-frozen, embedded mouse brain microbiopsy punches. We found that holding cryo-chamber and chuck temperatures at −11 °C improves flexibility of the fragile flash-frozen tissue, while maintaining adherence of embedded tissue to the sample mount, thus allowing for uninterrupted sectioning of alternating 100–300 µm sections for punching, and paired 20 µm sections for histology (Fig. 1a). This approach facilitates acquisition of both sections for microbiopsy punching and paired sections compatible with IHC staining and high-resolution microscopy. Cryopreservation of 100–300 µm/20 µm slide decks at −80 °C allows for long-term sample storage and the ability to test hypotheses by staining after analysis of the spatially resolved chromatin accessibility profiles; however, we note that sections stored for ~3 months result in an overall loss of quality in transcription start site (TSS) enrichment.

Microbiopsy punching of 100–300 µm sections performed within a cooled chamber ("Methods") allows for isolation of microscopic pellets of nuclei that readily dissociate in nuclear isolation buffer (NIB) after mechanical dissociation by trituration. We observed minimal loss after pelleting and washing nuclei, an important step for the removal of mitochondria, which can deplete the available pool of transposase because of the high transposition efficiency into mitochondrial DNA[24]. Nuclei isolation, as measured by nuclei per cubic micron, was more efficient for volumetrically smaller punches (Fig. 1b). This implies that smaller punches dissociate more readily because of a higher surface area to volume ratio, thus higher resolution punches yield more nuclei, respective of volume.

### sciMAP-ATAC performance and quality assessment.
We applied these techniques to perform sciMAP-ATAC, where we tested four methods of punch dissociation ("Methods"). We

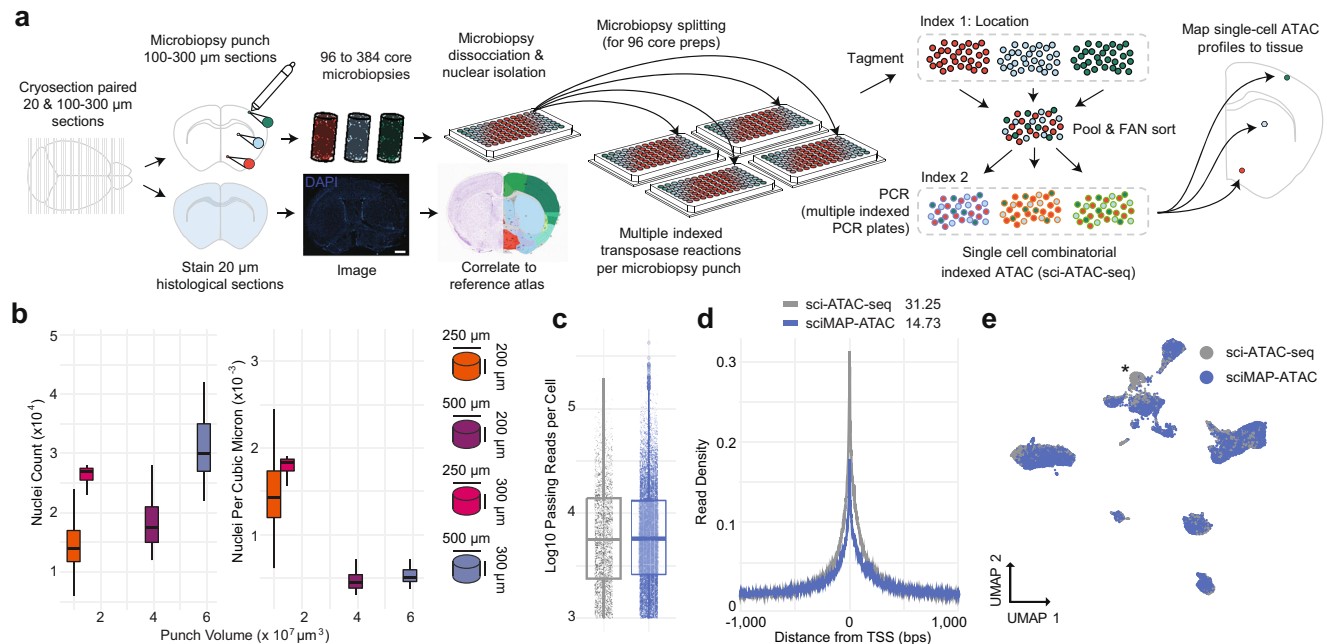

**Fig. 1 sciMAP-ATAC schematic and performance. a** sciMAP-ATAC workflow. Cryosectioning of alternating 20 μm (histological) and 100–300 μm (sciMAP-ATAC) slices are obtained. Thin (20 μm) slices are stained and imaged for use in spatial registration (scale bar, 1 mm) to a reference atlas (Allen Mouse Brain Atlas: http://atlas.brain-map.org/atlas?atlas=1&plate=100960312, ref. [25]). Thick (100–300 μm) slices are carried through high-density microbiopsy punching (100–500 μm diameter) in the cryostat chamber. Punches are placed directly into wells of a microwell plate for nuclei isolation, and washed prior to splitting into multiple wells for indexed transposition and the sci-ATAC-seq workflow. **b** Four punch volumes were assessed for nuclei yield using either a 250 or 500 μm diameter punch on a 200 or 300 μm thick section. Total nuclei isolated for each punch is shown on the left, and normalized for tissue voxel volume on the right, representing the efficiency of extraction from each punch, for punches with dimensions 250 ×200 μm ($n = 48$), 250 × 300 μm ($n = 15$), 500 × 200 μm ($n = 46$), and 500 × 300 μm ($n = 7$). Center line represents median, lower and upper hinges represent first and third quartiles, and whiskers extend from hinge to ±1.5 × IQR. **c** Passing reads per cell from sci-ATAC-seq ($n = 4102$ cells examined from a single mouse brain experiment) and sciMAP-ATAC ($n = 15,552$ cells examined from two independent mouse brain experiments), which are comparable at the level of depth sequenced. Center line represents median, lower and upper hinges represent first and third quartiles, whiskers extend from hinge to ±1.5 × IQR, individual cells represented as colored dots. **d** ATAC read signal at transcription start sites (TSSs) and surrounding base pairs (bps) for sci-ATAC-seq and sciMAP-ATAC. Enrichment for sci-ATAC-seq is greater than that of sciMAP-ATAC, likely due to increased processing time of isolated nuclei prior to transposition. **e** UMAP of sciMAP-ATAC and sci-ATAC-seq libraries from mouse brain group closely together. Asterisk indicates a population of 734 cells, derived from spinal cord, which was not sampled during microbiopsy punching. Source data are provided as a Source data file.

utilized a workflow similar to our established sci-ATAC-seq method, with each indexed transposition reaction performed on an individual punch, for a total of 384 transposition reactions, performed in four 96-well plates. Reactions were pooled and indexed nuclei were distributed via fluorescence-assisted nuclei sorting (FANS) to wells of four new 96-well plates for indexed real-time PCR, followed by pooling and sequencing. The resulting library produced 8011 cells passing filters, for an estimated doublet rate of 2.5% based on the total indexing space of 384 × 384 ("Methods"), and a mean of 12,052 passing reads per cell (unique reads, aligned to autosomes or X chromosome at q10 or higher; Supplementary Fig. 2a) at the depth sequenced and potential to reach 23,830 mean passing reads per cell with additional sequencing ("Methods"). This is comparable to the mean passing reads per cell from the whole brain sci-ATAC-seq library at 11,987 (projected mean passing reads of 24,672 and 32,029 for fresh and frozen preparations, respectively; Fig. 1c and Supplementary Fig. 2a). We observed a mean of 112 passing cells per punch. This could be increased if additional PCR plates were sorted, as the pool of indexed nuclei were not depleted during FANS. A comparison between the four dissociation methods enabled us to identify an optimal means of punch processing that produced the highest cell counts per punch with high-quality cell profiles (Methods; Supplementary Figs. 1b and 2a), which was used for all subsequent experiments. Across all sciMAP-ATAC datasets produced in this study on healthy mouse brain tissue, we

achieve a TSS enrichment of 14.73, within the "acceptable" range prescribed by ENCODE (10–15, mm10 RefSeq annotation) and just shy of "ideal" (>15). This is substantially below that of our sci-ATAC-seq preparation, with a TSS enrichment of 31.25; however, we note that an enrichment of more than double the "ideal" standard is exceptionally high ("Methods", Fig. 1d). In line with the lower TSS enrichment in sciMAP-ATAC, we also observed a reduction in the fraction of reads present in a mouse reference peak set (FRiS; "Methods"), with a mean ranging from 0.83 to 0.87, compared to 0.91 and 0.93 for sci-ATAC-seq (Supplementary Fig. 2b). Finally, we performed an integrated analysis across these preparations that revealed negligible batch effects (Fig. 1e and Supplementary Fig. 3a, b). We observed a single exception in the form of a population of cells present only in the nonspatial dataset which, upon inspection, were determined to be spinal cord derived interneurons (Supplementary Fig. 3c, d) and not present in coronal sections that were used in spatial experiments. Taken together, with improvements and validation on sample preparation, cryosectioning, nuclei isolation, and the general sci-ATAC-seq protocol, we generated a robust method to obtain the spatial information that we sought to test in a complex system.

**sciMAP-ATAC in the adult mouse somatosensory cortex.** To establish the ability of sciMAP-ATAC to characterize single cells

within a spatially organized tissue, we applied the technique to resolve murine cortical lamination within the primary somatosensory cortex (SSp). We harvested intact whole brain tissue from three wild-type C57/Bl6J adult male mice, flash-froze the tissue, and prepared whole brain slide decks of 200 μm microbiopsy slides each interspersed with three 20 μm histological slides. To orient sections to intact mouse brain, and to establish the quality of histological section prepared according to the sciMAP-ATAC protocol, we stained nuclei using DAPI and IHC stained for SATB2 to resolve cortical layers ("Methods", Fig. 2a). DAPI imaging was then matched to the adult mouse Allen Brain Reference Atlas[25], which enabled determination of the SSp location within adjacent sections for punch acquisition. SATB2 imaging demonstrated the quality of histological sections, across diverse fixation protocols (4% PFA postfixation for 10 min and 70% ethanol postfixation for 30 s) and generated a high signal-to-noise ratio canonical for SATB2 IHC staining[26] (Fig. 2b). Microbiopsy punches were then taken from three regions: (i) outer (L2–4) SSp cortical layers, (ii) inner (L5 and 6) SSp cortical layers, and (iii) throughout the striatum. The striatum is rich in glia and is absent of cortical glutamatergic neurons and cortical lamination. Therefore, the striatum punches served as a negative control for these features and also bolstered single-cell glial cell type identification. In total, 96 individual tissue punches were obtained, split evenly between the three categories over eight coronal sections spanning the SSp (Fig. 2a). After nuclei isolation, each well of the plate containing a single punch was split across four wells, resulting in four 96-well plates for subsequent indexed transposition, providing four tagmentation technical replicates for each punch. Posttransposition, nuclei were pooled and

distributed to two 96-well PCR plates for the second tier of indexing and then sequenced ("Methods").

We processed the raw sequence data ("Methods"), which resulted in 7779 cells passing quality filters (estimated doublet rate of 4.9%; "Methods"). Our mean passing reads per cell was 17,388, with a projected total passing mean reads per cell of 37,079 ("Methods"), a TSS enrichment ranging from 13.74 to 15.26, and nucleosomal banding present in the library insert size distribution (Supplementary Fig. 2a–d). A median of 81 single-cell profiles was obtained per punch, with little bias for punch target region or section (Supplementary Fig. 2d). Subsequent peak calling, topic modeling, and dimensionality reduction ("Methods") revealed cell groupings that were either mixed between the three regional categories or highly enriched for cells derived from the cortex, which was further divided by outer versus inner punch location (Fig. 2c, Supplementary Fig. 3e, f, and Supplementary Data 2). Overlay of spatial data on the UMAP projection fits with our expectation that glutamatergic (excitatory) neurons are cortex exclusive, displaying an absence of punch-to-punch cross talk or contamination. In addition, these cells were integrated with prior sciMAP-ATAC, and sci-ATAC-seq experiments where excitatory neuron clusters were also dominated by cortex-derived punches, with a shared spatial bias between upper and lower punch positions. This demonstrates that spatial datasets can be integrated with nonspatial datasets to provide additional spatial information to those datasets, using label transfer or other analysis techniques (Supplementary Fig. 3a, b).

We identified eleven clusters over eight broad cell type groups corresponding to glutamatergic neurons, GABAergic (inhibitory) neurons, GABAergic medium spiny neurons (MSNs; also referred

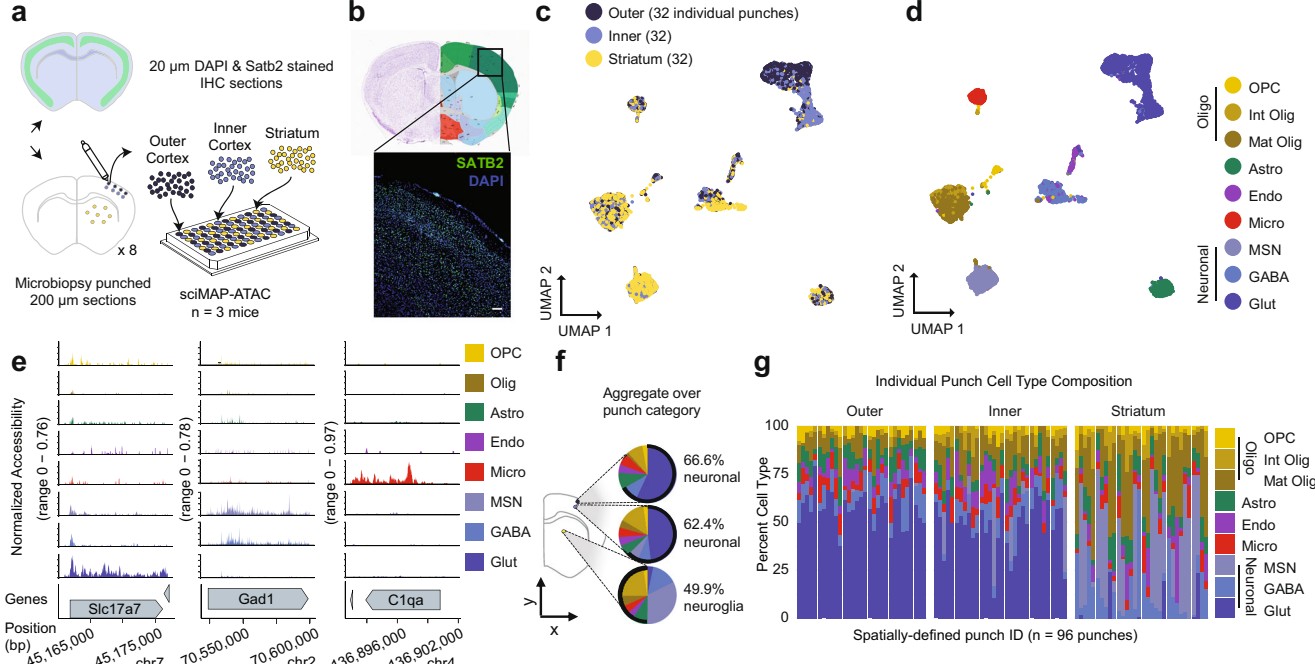

**Fig. 2 sciMAP-ATAC reveals spatially distinct cell type composition in the mouse somatosensory cortex. a** Experiment schematic of sciMAP-ATAC in the mouse somatosensory cortex. **b** DAPI and SATB2 staining of SSp cortex from sciMAP-ATAC histological section (scale bar, 50 μm) in reference to matched reference atlas image (Allen Mouse Brain Atlas: http://atlas.brain-map.org/atlas?atlas=1&plate=100960312, ref. [25]). **c** UMAP of 7779 cells colored by punch location category. Each category contains cells from 32 spatially distinct tissue punches. **d** UMAP as in **c**, colored by cell type (OPC oligodendrocyte precursor cells, Int Olig intermediate oligodendrocytes, Mat Olig mature oligodendrocytes, Astro astrocytes, Endo endothelia, Micro microglia, MSN medium spiny neurons, GABA GABAergic (inhibitory) neurons, Glut glutamatergic (excitatory) neurons). **e** ATAC-seq profiles for cells aggregated by cell type for marker genes; colored by cell type as in **d**. **f** Aggregate cell type composition over punches belonging to the broad region categories; colored by cell type as in **d**. **g** Cell type composition for each of the 96 individual punches split by broad region category; colored by cell type as in **d**. Source data are provided as a Source data file.

to as spiny projection neurons (SPNs)), oligodendrocyte precursor cells (OPCs), newly formed or intermediate oligodendrocytes, mature oligodendrocytes, astrocytes, microglia, and endothelial cells based on the chromatin accessibility signature of regulatory elements proximal to marker genes ("Methods"; Fig. 2d, e and Supplementary Data 1). GABAergic neurons subdivide into non-layer-specific cortical GABAergic neurons and striatum-derived MSNs. In contrast, glutamatergic neurons separate along the dorsal–ventral axis, as determined by punch position. This recapitulates known neuronal cell state biology, where glutamatergic pyramidal neurons express cortical layer(s)-specific markers that define the spatially defined cortical layers. Within the SSp-derived cells, we observed 66.6%, 62.4%, and 49.9% of cells corresponding to neurons in the inner cortex, outer cortex, and striatum, respectively. These equate to glia to neuron ratios (GNRs) of 0.50, 0.60, and 1.00 from inner cortex, outer cortex, and striatum, respectively, which correspond to previously reported mouse cerebral cortex and striatum GNRs of 0.66 and 0.97, respectively[27]. In addition to coarse cell type characterization across the major punch categories, we determined cell type composition for each individual spatially resolved punch (Fig. 2g). For cortical punches, little variance was observed within the outer and inner punch categories; however, we did observe increased variability in the proportion of MSNs in the striatum punches, ranging between 2.78% and 72.64%, suggesting a non-even

distribution of these cells, which is confirmed by MSN cell type marker, *Drd1*, in situ hybridization in adult C57BL/6J striatum (Allen Mouse Brain Atlas)[25].

**Analysis of individual punch sciMAP-ATAC profiles and spatial comparisons.** We next characterized the single-cell ATAC profiles produced from a single tissue punch. We isolated cell profiles that were from punch F5 ($n = 90$ cells), an inner cortex punch, and performed the same analysis as above using the set of peaks called on the full dataset. This produced a set of topic weights that contained a clear structure and were associated with specific cell types (Fig. 3a). This was also clear in the UMAP projection, with three primary clusters of cells identified (Fig. 3b). Two of these groups were dominated by one cell type, including glutamatergic neurons and GABAergic neurons, with the third group comprised predominantly of glial cell types.

We then took the examination of this individual punch further by performing all aspects of the analysis, including peak calling, on only the cell profiles present in punch F5. From those 90 cells, we were able to call 8460 peaks which were sufficient to perform topic modeling and UMAP visualization, and identify two distinct clusters: one comprised of glutamatergic neurons, and the second containing all other cell types, based on the cell type identities established in the analysis of the full dataset (Fig. 3c, d). A

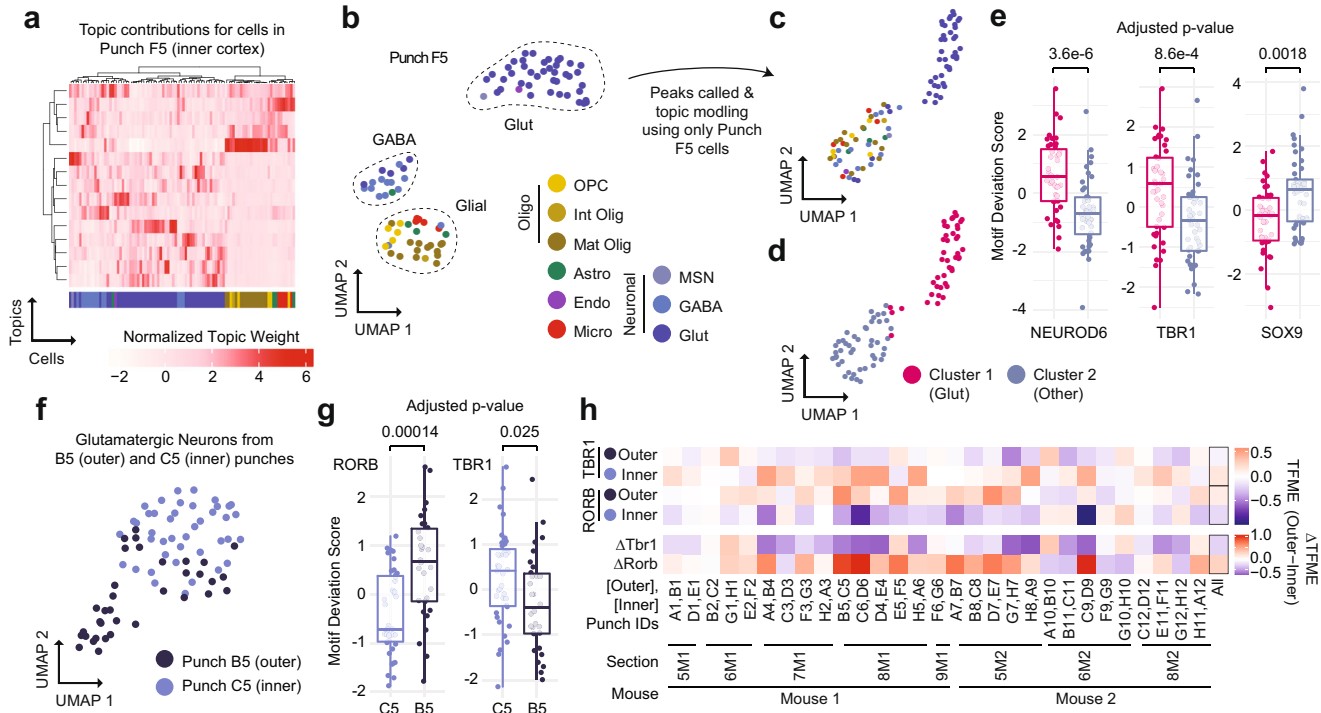

**Fig. 3 sciMAP-ATAC enables the analysis and comparison of cells and cell types from individual spatial positions. a** Topic weight matrix for cells present only in a single punch (F5, inner cortex punch), annotated by cell type (bottom); colored by cell type from the full dataset (Fig. 2d). **b** UMAP of cells from punch F5 showing spatially distinct groupings for cell type; colored by cell type from the full dataset (Fig. 2d). **c** Isolated analysis of cells from Punch F5 for peak calling, topic modeling, and visualized via UMAP; colored by cell type from the full dataset (Fig. 2d). **d** Two major clusters identified from the isolated analysis of punch F5 (Glut glutamatergic (excitatory) neurons). **e** Transcription factor motif enrichments for the isolated analysis of punch F5, indicating that cluster 1 ($n = 44$ cells) is made up of glutamatergic neurons and cluster 2 ($n = 45$ cells) is made up of other cell types. Center line represents median, lower, and upper hinges represent first and third quartiles, whiskers extend from hinge to ±1.5 × IQR, individual cells represented as colored dots. **f** UMAP of all glutamatergic neuron cells from two adjacent punches (C5, inner cortex, and B5, outer cortex) after topic modeling on the isolated cell profiles. **g** Transcription factor motif enrichments for glutamatergic cells from adjacent punches from inner cortex ($n = 39$ cells) and outer cortex ($n = 30$ cells) shown in **f**; colored by individual punch as in **f**. Two-sided Mann–Whitney U test with Bonferroni–Holm correction. Center line represents median, lower and upper hinges represent first and third quartiles, whiskers extend from hinge to ±1.5 × IQR, individual cells represented as colored dots. **h** Motif enrichments across glutamatergic neurons across all punch pairs. TFME transcription factor motif enrichment. Source data are provided as a Source data file.

comparison of global motif enrichment between the two clusters revealed elevated NEUROD6 and TBR1, and depleted SOX9 motif accessibility in the cluster comprised of glutamatergic neurons, suggesting very coarse cell type class assignment can be performed on data from a single punch analyzed in isolation (Fig. 3e). Further resolution of cell types on such a small number of cells, especially without leveraging larger peak sets, is not likely feasible simply due to the low abundance of certain cell types—for example, there was only one endothelial cell present in punch F5. However, it is unlikely that individual punches would be profiled alone in an experiment and the throughput provided in sciMAP-ATAC enables identification of low-abundance cell types in the aggregate dataset, which can be used when performing analysis on individual punch positions.

Finally, we explored whether we could identify and characterize spatially distinct chromatin properties from a single cell type present within two adjacent punches. We isolated cells that were identified as glutamatergic neurons in two punches, C5 (inner cortex) and B5 (outer cortex), that were immediately adjacent with 83 and 65 total cells, and 42 and 35 glutamatergic cells, respectively. Similar to the single punch analysis, we produced a counts matrix including only these cells and used the full set of peaks to perform topic analysis and visualization using UMAP, which showed clear separation between the two locations (Fig. 3f). We then assessed global motif accessibility, which revealed clear enrichment for motifs associated with upper or lower cortical layers, including RORB, enriched in the outer cortex, and TBR1, enriched in the inner cortex (Fig. 3g). To systematically assess this spatial TF motif enrichment (TFME), we applied this same analysis to the glutamatergic cell populations identified in every pair of inner and outer cortical punches. This produced a consistent pattern with very few punch pairs deviating from the expected enrichment pattern (Fig. 3h).

**Spatial trajectories of single-cell ATAC-seq in the human cortex.** With the ability to probe spatial single-cell chromatin accessibility established in the mouse cortical lamination experiment, we next deployed sciMAP-ATAC on human brain tissue to profile lamination in the adult primary visual cortex (VISp) using an equivalent voxel-diameter resolution of 215 cubic microns. Samples of human VISp tissue were obtained from an adult (60-year male) with no known neurodegenerative disorders at 5.5 hours postmortem. Samples were oriented and flash-frozen in TFM prior to storage at −80 °C. The sample was cryosectioned using the same alternating thick (200 μm) and thin (20 μm) pattern, as previously described. We designed and implemented a 250 μm diameter punch schematic across three adjacent 200 μm sections to produce 21 distinct trajectories comprised of eight punches spanning the cortex, with an additional 20 punches distributed in the subcortical white matter for a total of 188 spatially mapped tissue punches (Fig. 4a, b). In total, 4547 cells passed quality filters with a mean of 30,212 reads per cell (estimated mean of 98,274 passing reads per cell with additional sequencing; "Methods", Supplementary Figs. 2a and 4a), a mean TSS enrichment of 15.80—more than twice the "ideal" ENCODE standard for bulk ATAC-seq datasets (>7, GRCh38 RefSeq annotation), a FRiS of 0.45 using a human reference dataset[28], and prominent nucleosomal banding ("Methods", Supplementary Fig. 2b, c, e).

Cell profiles were generated as described in prior experiments, which resulted in six distinct clusters representing the major cell types (Fig. 4c, d). Similar to the murine cortex, glutamatergic neurons exhibited the most distinct spatial patterning with a clear gradient spanning cortical trajectories (Fig. 4a–c), which was also determined to be the most significant (Moran's $I$ test Bonferroni corrected $p$ value $= 0.87 \times 10^{-4}$, "Methods", Supplementary Table 1). Further subclustering of GABAergic interneurons revealed

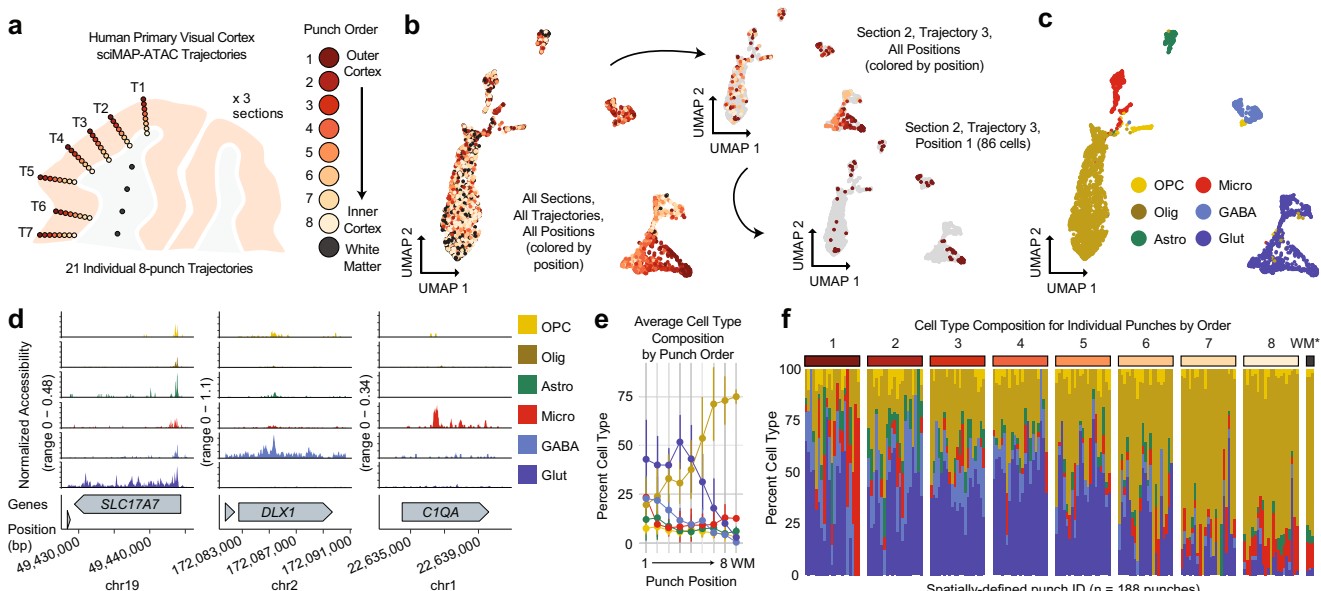

**Fig. 4 sciMAP-ATAC trajectories through the human primary visual cortex. a** sciMAP-ATAC punching schematic showing one of three adjacent sections from one individual. A total of 21 eight-punch trajectories (T) spanning the cortex were produced. **b** UMAP of cells colored by position within their respective trajectory as in **a**. Top right shows the same UMAP with all cells grayed out with the exception of cells from the third trajectory from section 2. Bottom right shows all cells grayed out with the exception of cells from a single punch; the outermost cortical position (1) from the third trajectory of the second section. **c** UMAP as in **b** colored by cell type (OPC oligodendrocyte precursor cells, Olig oligodendrocytes, Astro astrocytes, Micro microglia, GABA GABAergic (inhibitory) neurons, Glut glutamatergic (excitatory) neurons). **d** ATAC-seq profiles for cells aggregated by cell type for marker genes; colored by cell type as in **c**. **e** Aggregate cell type composition across the 21 trajectories (n = 4547 cells over 188 independent punches); colored by cell type as in **d**. Data are presented as mean values ± SD. **f** Cell type composition for each of the 188 individual punches split by trajectory position. Punches from the WM indicated by an asterisk are aggregated by section. Colored by cell type as in **d**. Source data are provided as a Source data file.

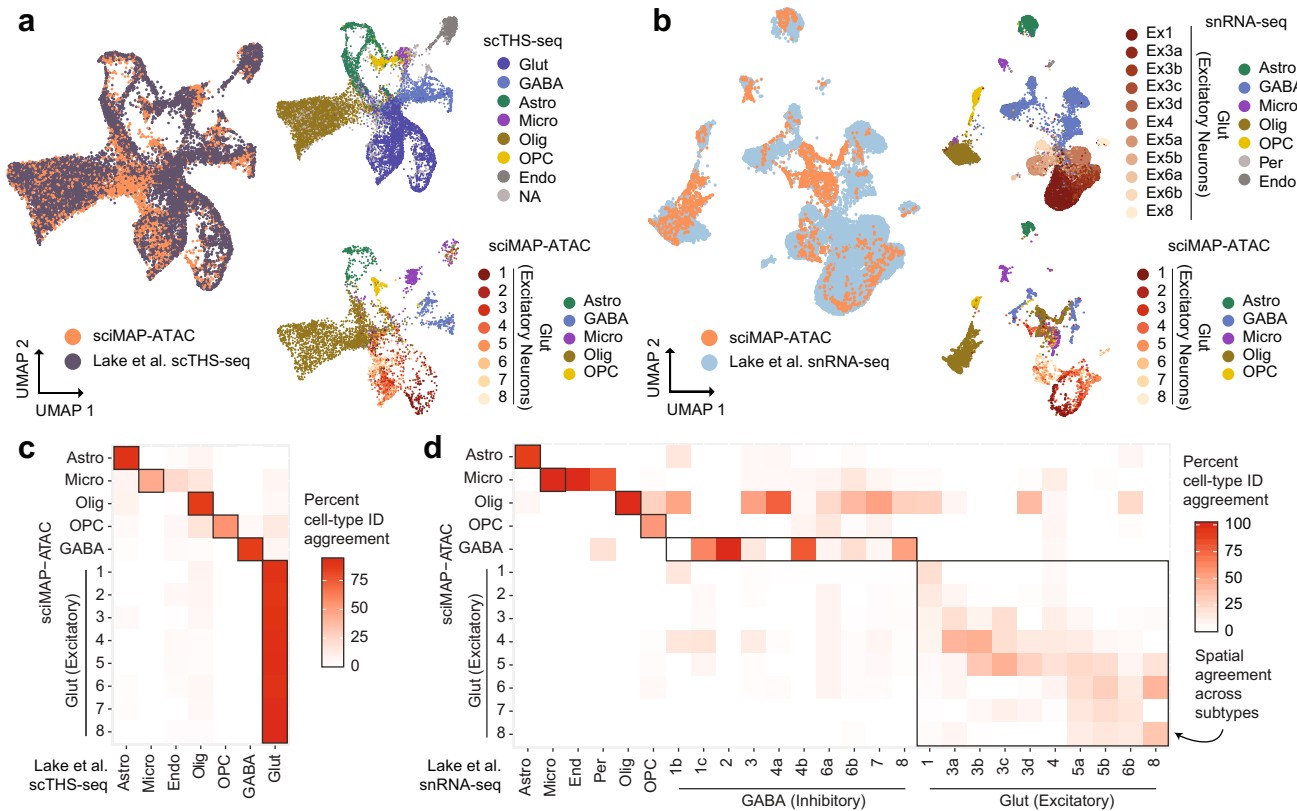

**Fig. 5 Integration of sciMAP-ATAC with snRNA-seq and scTHS-seq human VISp datasets. a** Co-embedding of sciMAP-ATAC and scTHS-seq cell profiles from Lake et al.[29] using Signac[75] in a joint UMAP. Top right shows only scTHS-seq cells colored by cell type identified in Lake et al.[29] and bottom shows sciMAP-ATAC cells colored by our called cell types as in Fig. 4c, except for glutamatergic neurons which are colored by spatial positions 1–8 (Glut glutamatergic (excitatory) neurons, GABA GABAergic (inhibitory) neurons, Astro astrocytes, Micro microglia, Olig oligodendrocytes, OPC oligodendrocyte precursor cells, Endo endothelial cells, NA not applicable—no cell type provided). **b** Co-embedding of sciMAP-ATAC and snRNA-seq transcriptional profiles from Lake et al.[29] using Signac. Top right shows only snRNA-seq cells. Abbreviations as in **a**, but with the addition of Per = pericytes, and glutamatergic (excitatory) neurons (Ex) are colored by subtype identified in Lake et al.[29]. Bottom right shows only sciMAP-ATAC cells, with glutamatergic neurons colored by spatial position 1–8. **c** Confusion matrix representing the percent agreement in predicting the cell type of a cell from one dataset using the other between sciMAP-ATAC and scTHS-seq cells. **d** As in **c**, but between sciMAP-ATAC and snRNA-seq. Spatial agreement between excitatory neuron subtypes identified in the snRNA-seq data correspond to the spatial positioning of cells within our sciMAP-ATAC dataset. Source data are provided as a Source data file.

minimal spatial bias across four distinct subtypes comprised of two MGE-derived and two CGE-derived clusters (Supplementary Fig. 4b–e). Each of the 21 individual trajectories through the cortex produced similar distributions of cells through UMAP projections with a lack of glutamatergic neurons present in the punches obtained from subcortical white matter (Supplementary Data 2). Our astrocyte to neuron ratio (0.15:1) was low, yet comparable to previously published snRNA-seq of the human VISp (0.12:1)[29]. Average cell type composition along these trajectories revealed the expected pattern of an increased proportion of oligodendrocytes and decreased glutamatergic neuron abundance, as the trajectory approached or entered the subcortical white matter region (Fig. 4e). Individual punches largely matched the corresponding average position profile (1–8, WM), with higher variability at the first punch where some trajectories overlapped the pial surface of the cortex (Fig. 4f).

**Integration of sciMAP-ATAC with scTHS-seq and snRNA-seq reveals epigenetic spatial patterning concordant with transcriptional neuronal subtypes.** Previously, Lake et al. produced single-cell transposase hypersensitivity (scTHS-seq, an assay for chromatin accessibility similar to ATAC-seq) and single-nucleus RNA-seq from the human VISp[29]. We integrated our sciMAP-

ATAC dataset with each of these using Seurat[30] and visualized the joint UMAP projections with cell type information, along with the positional breakdown of glutamatergic neurons (Fig. 5a, b). The joint manifold for each integration largely agreed, with the exception of a population of cells in our sciMAP-ATAC dataset that did not co-embed with any cell types present in the snRNA-seq dataset. These cells represent all of the cell types called within the sciMAP-ATAC dataset, and cluster clearly with their cell types in the sciMAP-ATAC analysis on its own, suggesting that it may be an effect of the gene activity score intermediate that is used for co-embedding with ATAC-based data ("Methods").

To directly assess the performance of the dataset integration, we used the joint manifold to perform cell type label transfer, effectively using one assay's cell type identities to predict the other's, and compared the overlap in the form of a confusion matrix. For the scTHS-seq integration, the top concordance was between the two corresponding cell types in nearly every case, including across all eight of the spatial glutamatergic neuron cell sets within the sciMAP-ATAC dataset that all corresponded to the single glutamatergic cell type in the scTHS-seq dataset (Fig. 5c). One exception was the association of a subset of microglia within the sciMAP-ATAC dataset with the endothelial cell population identified in the scTHS-seq dataset, which is a population we did not define. This suggests that a portion of our

cells identified as microglia are likely endothelial cells. Integration with snRNA-seq data also produced concordance for the majority of cell types (Fig. 5d), with the exception of a group of cells spanning all cell types that did not co-embed as cleanly, and thus project into the center of the UMAP. The snRNA-seq data provided in Lake et al. includes a more granular breakdown of glutamatergic neurons when compared to the single classification provided for scTHS-seq cells. Within the confusion matrix where cell types were predicted across modalities, we observed a clear spatial progression that corresponded to the subtypes of glutamatergic neurons identified by snRNA-seq, which Lake et al. previously identified as being enriched for layer-specific transcripts. The concordance between these subtypes and our spatial assignments confirms that sciMAP-ATAC spatially registers biological features of single cells from structured tissue.

**Spatial excitatory neuron epigenetic patterning at the individual trajectory level**. Using our cell type assignments, we isolated all human VISp glutamatergic neurons and split them by position along their respective trajectories (Fig. 6a and Supplementary Data 3). We examined ATAC signal at layer-specific marker genes broken down by each spatially distinct category, which revealed increased accessibility at genes associated with outer cortical layers within the outer cortical punches and vice versa (Fig. 6b). We next selected all cells from the centermost trajectory of section 1 (T1.4, $n = 358$ cells) and performed an isolated

analysis using peaks called on the full dataset for topic analysis, cluster identification, and visualization with UMAP (Fig. 6c and Supplementary Fig. 4f, i). Clear separation was observed between major cell types across six clusters, with two distinct clusters of oligodendrocytes, two clusters of glutamatergic neurons, one cluster comprised of GABAergic neurons, and finally, a cluster made up of all other cell types (astrocytes, endothelial, and OPCs). When performing the analysis in isolation using only T1.4 cells for peak calling, we identified 16,493 peaks that were used for subsequent analysis to produce four clusters with notably less cell type separation than when leveraging the set of peaks from the full dataset (Supplementary Fig. 4j, l). The first cluster was comprised of both glutamatergic and GABAergic neurons, the second was primarily oligodendrocytes, the third included oligodendrocytes, as well as the majority of cells from all other nonneuronal cell types, with the fourth cluster comprised of only a handful of cells with no dominant cell type. In line with the previous assessment of a single punch from the mouse SSp, cell type separation can be distinct for major cell types when leveraging larger peak sets than the limited number that can be called on small cell count datasets. This supports the assertion that computational improvements to enable peak calling on low cell count datasets can substantially boost analytical power[31].

Finally, we isolated only cells determined to be glutamatergic neurons based on the full dataset cell type assignment within Trajectory 1.4 ($n = 121$ cells). We assessed these cells again using the full peak set through the same analysis workflow ("Methods").

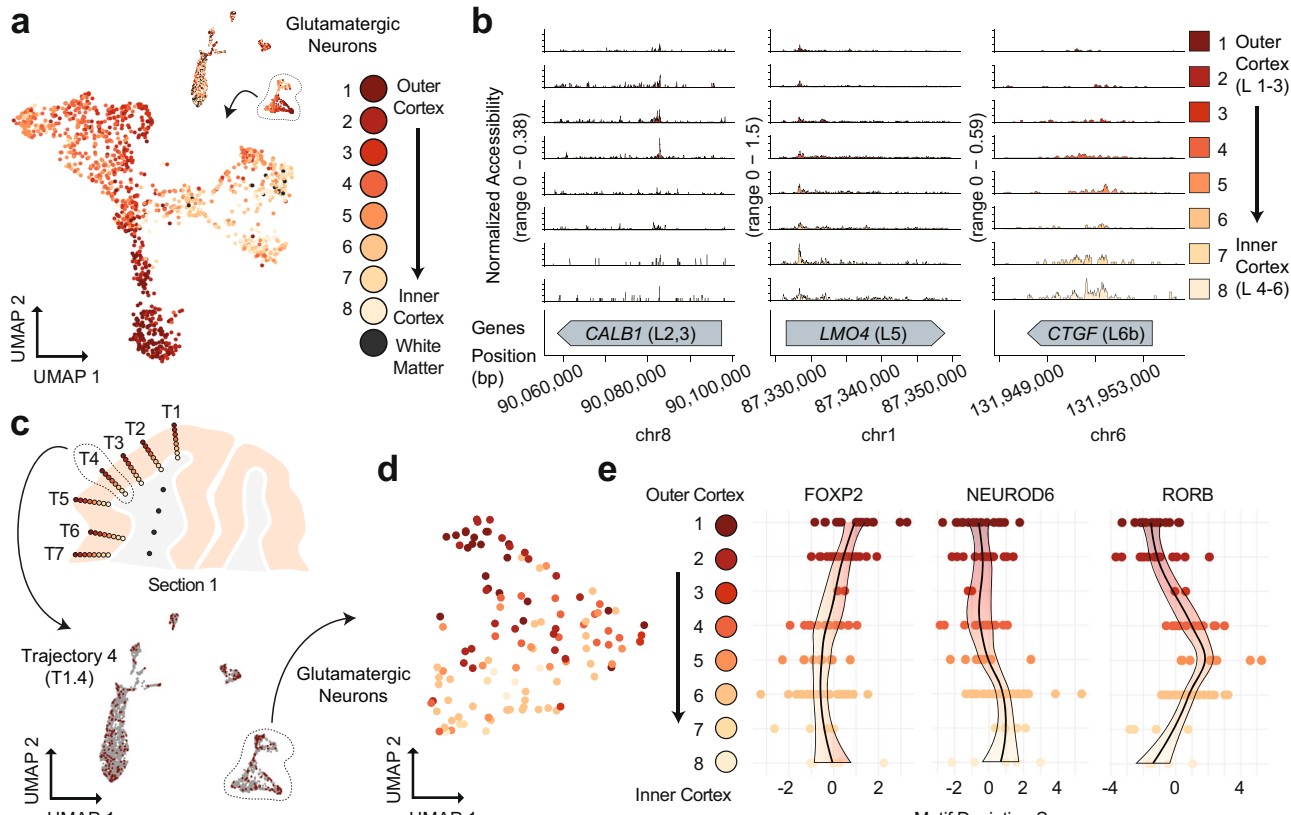

**Fig. 6 sciMAP-ATAC shows spatial epigenetic patterns of glutamatergic neurons. a** Isolation and UMAP visualization of human VISp glutamatergic neurons from all cells (top right), colored by punch position. An interactive, three-dimensional UMAP embedding is available as Supplementary Data 4. **b** ATAC-seq profiles for glutamatergic neurons along trajectory positions for layer (L)-specific marker genes CALB1 (layers 2 and 3), LMO4 (layer 5), and CTGF (layer 6b); colored by punch position as in **a**. **c** Cells from section 1, Trajectory 4 (T1.4, top) are shown in color on the UMAP of all cells, with other cells shown in gray (bottom); colored by position as in **a**. **d** UMAP of glutamatergic neurons from Trajectory 1.4 after topic modeling on the isolated cells; colored by position along the trajectory as in **a**. **e** DNA-binding motif enrichment for layer-specific factors for Trajectory 1.4 shown in **d**, with cells split by their positions along the trajectory. Source data are provided as a Source data file.

As in the UMAP projections on cells from the full experiment, these cells were positioned along a gradient that reflected their position along the trajectory (Fig. 6d). We then assessed the global accessibility of DNA-binding motifs that captured spatially distinct enrichments through the trajectory reflecting the expected pattern of transcription factor (TF) activities through cortical layers (Fig. 6e). This included enrichment for FOXP2 motif accessibility in the outer cortical layers, slightly increased accessibility for NEUROD6 toward the inner cortex, and increased the accessibility for RORB motifs in punches 4–6 along the trajectory, corresponding to canonical cortical layer 4 RORB expression. Taken together, sciMAP-ATAC is capable of producing high-quality single-cell ATAC-seq profiles from postmortem human tissue with a spatial resolution capable of identifying the major components of cortical lamination, with the capability to characterize a single spatial trajectory through the cortex.

**sciMAP-ATAC in a mouse model of cerebral ischemia.** Cerebral ischemia produces a complex spatially progressive phenotype with extensive tissue alterations and shifts in cell type abundance and epigenetic states[32–37]. Cerebral ischemic infarction induces gliosis, a process in which glia in the surrounding tissue enter reactive states that are potentially aimed at restoring tissue homeostatis, but can involve the loss of normal function (or adoption of a damaging function) and form a glial scar. Many components involved in the ischemic cascade are well studied, including factors that promote postischemic inflammation (e.g., IRF1, NF-kB, ATF2, STAT3, EGR1, and CEBPB), and prevent postischemic inflammation and neuronal damage (e.g., HIF-1,

CREB, C-FOS, PPARα, PPARγ, and P53)[38]. Reactive gliosis can be characterized by increased GFAP expression in astrocytes and increased IBA1 in microglia. Myelination depletion is a hallmark of cerebral ischemic injury, due to acute oligodendrocyte cell death and impaired OPC differentiation[39,40]. Far less is known, however, about glial cell state transitions in the area surrounding ischemic infarction in the brain. We reasoned that our sciMAP-ATAC technology could reveal, with cell type and spatial specificity, the epigenetic alterations that occur to accompany and/or drive the ischemic cascade and postischemic pathology.

To accomplish this, we used a transient middle cerebral artery occlusion (MCAO) mouse model of ischemic injury with reperfusion ("Methods"; Fig. 7a). Each ischemic ($n = 2$ animals) and naive ($n = 3$) brain was flash-frozen 3 days after surgery, embedded in TFM, sectioned, alternating between 200 μm for sciMAP-ATAC and 20 μm for IHC for IBA1 (microglia), GFAP (astrocytes, Fig. 7b), and counterstained using DAPI. We used these images to define the infarct area by absence of GFAP-positive astrocytes, while being surrounded by reactive astrocytes exhibiting increased GFAP signal at the infarct border (Supplementary Fig. 5a). We next defined two axes for targeting the sciMAP-ATAC punches, the first progressing from the pial surface of the cortex to the striatum, all within the infarct core (punch position axis 1–4), and the second progressing from the infarct core toward the infarct border (punch position axis 5–8). GFAP immunolabeling was absent in the infarct core (punch positions 5–7) but increased at the infarct border in punch position 8, recapitulating known features of glial scar formation surrounding the infarct area. We then performed sciMAP-ATAC on the 200 μm sections along each axis to produce 5081 cells with

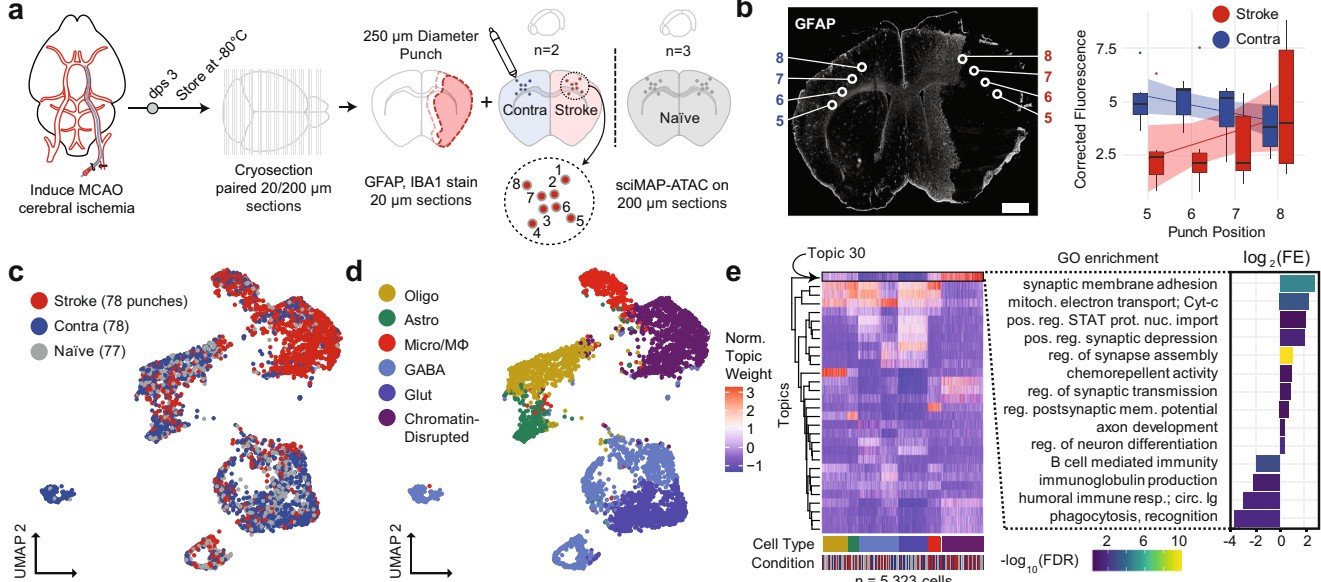

**Fig. 7 sciMAP-ATAC applied to a mouse model of ischemic injury. a** Experimental design using a mouse MCAO model of ischemic injury. Mice were sacrificed 3 days post surgery (dps) and brains flash-frozen in TFM. Alternating thin (20 μm) and thick (200 μm) sections were processed using IHC to define infarction (red outline) and peri-infarct area (pink outline) and sciMAP-ATAC punching schematic, respectively. **b** GFAP IHC of a 20 μm coronal section of an ischemic mouse brain. Punch positions along the 5–8 axis (core-to-border) are indicated. Background corrected GFAP fluorescence along the 5–8 axis is shown to the right for stroke and contralateral hemispheres ($n = 10$). Data are presented as linear fitted model ± SEM; boxplot center line represents median, lower and upper hinges represent first and third quartiles, and whiskers extend from hinge to ±1.5 × IQR, (scale bar, 1 mm) **c** UMAP of cells colored by the three conditions. **d** UMAP as in **c**, colored by clusters assigned to cell type (Olig oligodendrocytes, Astro astrocytes, Micro/MΦ microglia/macrophage, GABA GABAergic (inhibitory) neurons, Glut glutamatergic (excitatory) neurons). **e** Cell × topic matrix colored by normalized topic weights, as in **c**, **d** and annotated by conditions and cell type as given at the bottom reveals substantially divergent topic weighting in cells from the stroke punches (left). Topic 30, enriched specifically in the stroke cells belonging to the chromatin-disrupted cluster, has peaks enriched for ontologies associated with ischemic injury with reperfusion. Colored by $-\log_{10}$ false discovery rate (FDR) Q-value, height by $\log_2$ fold enrichment (right). Source data are provided as a Source data file.

a mean passing reads per cell of 33,832 (estimated mean passing reads per cell of 225,670 with further sequencing) and a mean of 26.6 high-quality cell profiles per punch (Supplementary Figs. 2a, f and 5b). TSS enrichment for this preparation was notably lower than previous preparations ranging from 5.05 (stroke hemisphere) to 7.50 (naive brain), which we suspect is due to several factors (Supplementarya, Fig. 2e). The first is that the stroke hemisphere contained many dead or dying cells that exhibit reduced ATAC signal, which we describe in more detail below, and the second is that these sections were stored for >3 months prior to sciMAP-ATAC processing, suggesting that long-term storage of sections may result in a reduction in data quality. Despite the reduced TSS enrichment and comparably lower FRiS (0.79–0.82; Supplementary Fig. 2b), we called 140,772 accessible genomic loci that were used in subsequent analysis.

We performed topic modeling, followed by clustering, cell type identification, and visualization of the cell × topic matrix (Fig. 7c–e), which revealed comparable cell type proportions across biological samples with exceptions for microglia/macrophages and a chromatin-disrupted cluster that were highly enriched within the infarct. We profiled cell type proportions along both of the axes (Supplementary Fig. 5c); however, the pial to striatum axis (punch positions 1–4) in stroke hemisphere samples is completely within the infarct core. In contrast, the infarct core-to-border axis (punch positions 5–8) progresses from the center of the infarct to the glial scar along the infarct border, capturing a transition zone of reactive gliosis, and is the spatial trajectory that we focus on in our subsequent analysis.

Along this progression, we found that the stroke hemisphere had diminished neural cell types (depletion of glutamatergic and GABAergic neurons, oligodendrocytes, and astrocytes), as well as a progressive increase in cells within a cluster exhibiting globally disrupted chromatin structure up to punch position 7 and a drop at punch position 8 upon entering the infarct border (Supplementary Fig. 5d). This state is predominantly characterized by globally increased chromatin accessibility, with a decrease in TSS enrichment, a decrease in FRiS, and an increase in reads falling within distal intergenic regions, which is likely caused by cell

death (Supplementary Fig. 5e, f). In addition to the global effects on chromatin structure, the chromatin-disrupted cell population also showed strong enrichment in one of the topics (Topic 30; Fig. 7e, left). A gene ontology (GO) enrichment analysis of the peaks that define topic 30 revealed that cells within the ischemic hemisphere undergo a chromatin state shift as a result of the ischemic cascade, which leads to enrichment for processes canonically associated with ischemia (Fig. 7e, right). Most notably, positive regulation of synaptic membrane adhesion, synaptic depression, assembly, transmission, and membrane potential were all enriched in ischemia-derived cells, indicating that CNS synaptogenesis is upregulated in a subset of cells 3 days post ischemia[41,42]. In addition, while the percentage of microglia increased in the stroke condition (13.2%) as compared to contralateral (6.7%) and naive (4.3%), depletion of immune response processes (B-cell-mediated immunity, humoral immune response mediated by circulating immunoglobulins) were seen in ischemia-derived cells. This recapitulates previous findings that acute ischemic immune response is followed by poststroke immunodepression and dysregulation[43,44].

**Spatially progressive chromatin features in cerebral ischemia.** To directly characterize the relationship between space and epigenetic state in cerebral ischemia, we assessed TF DNA-binding motif enrichments for each cell and performed a regression for all cells across the infarct core-to-border axis (punch positions 5–8) in the stroke and contralateral hemispheres. We used the difference between linear model coefficients for paired affected (stroke) and unaffected (contralateral) hemispheres along with the significance of the hemisphere motif enrichment differences to identify TFs that undergo spatially progressive regulatory changes ("Methods"). In total, we identified 95 TF motifs that were significantly altered with a spatial component, many of which have been previously reported as key factors identified in cerebral ischemia (Fig. 8a, b). KLF9, a member of the Kruppel-like factor family, demonstrated the most significant increase in accessibility with proximity to the peri-infarct area. The 17 KLF family TFs are

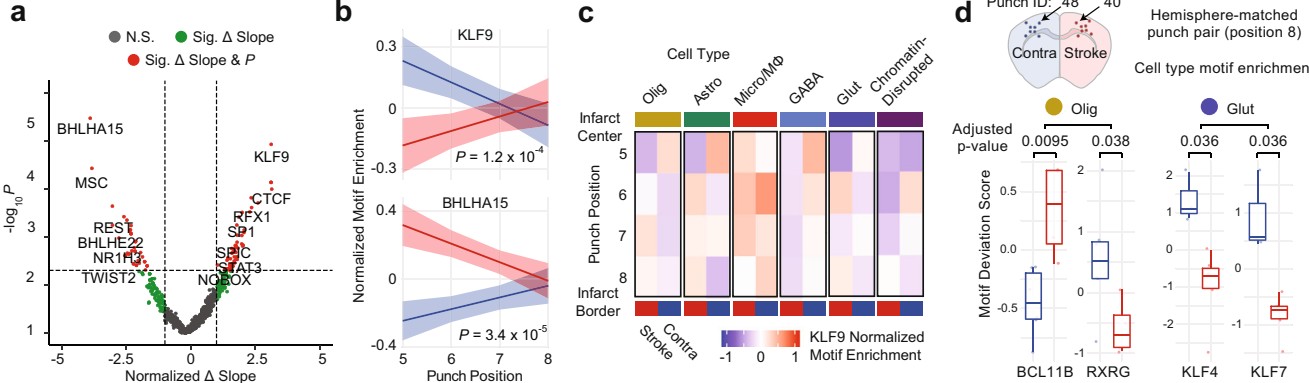

**Fig. 8 Spatially progressive epigenetic remodeling in ischemic injury. a** Volcano plot of Z-scored transcription factor (TF) motif enrichment slope change across punch positions 5–8 ($\Delta$slope = $slope_{stroke}$ − $slope_{contralateral}$) by −log10 p value of the two-way ANOVA from the interaction of TF motif enrichment per punch by condition (stroke, contralateral) without multiple comparison correction. Colored by significance (N.S. not significant, Sig. $\Delta$ slope = significant change in slope, Sig. $\Delta$ Slope and p = significant change in slope and significant p value). **b** Top hits for significantly different changes in TF motif enrichment over space as compared between stroke (red) and contralateral (blue); KLF9 (top) and BHLHA15 (bottom). −Log10 p value of the two-way ANOVA from the interaction of TF motif enrichment per punch by condition without multiple comparison correction. Data are presented as linear fitted model ± SEM. **c** KLF9 TF motif enrichment over space reveals cell type contribution to KLF9 enrichment from infarct core to peri-infarct area. Cell types as defined in Fig. 7d. **d** Comparison of TF motif enrichment at the infarct border (punch position 8) between stroke (punch 40) and contralateral (punch 48) single-cell profiles. Oligodendrocyte (Olig) TF motif enrichment shown for BCL11B and RXRG for punch 40 (n = 4 cells) and punch 48 (n = 6 cells). Glutamatergic neuron (Glut) TF motif enrichment shown for KLF4 and KLF7 for punch 40 (n = 5 cells) and punch 48 (n = 3 cells). Two-sided Mann–Whitney U test with Bonferroni–Holm correction. Center line represents median, lower and upper hinges represent first and third quartiles, whiskers extend from hinge to ±1.5 × IQR, individual cells represented as colored dots. Source data are provided as a Source data file.

key factors in neuronal development, plasticity, and axon regeneration and are ubiquitously expressed in the CNS. Several KLF family members, namely KLF2, 4, 5, 6, and 11, have been specifically linked to cerebral ischemia pathogenesis[45,46]. Notably, KLF2 and KLF11 have been shown to contribute to the protection of the blood–brain barrier in cerebral ischemia[47–49]. However, as DNA-binding motifs within the KLF family are similar, members of the KLF family other than KLF9 may be driving this motif accessibility change. Finally, we assessed the accessibility of individual elements and identified 73 accessible chromatin sites that varied significantly through the 5–8 axis of spatial progression ("Methods"; Supplementary Fig. 5g).

We next explored the cell type specificity of the KLF9 motif accessibility changes (Fig. 8c). In the stroke hemisphere chromatin-disrupted cell subset, we observed a reduction in KLF9 motif accessibility in all punch positions except punch position 8, at the infarct border, with all cell types other than microglia showing a reduction in accessibility at the center of the infarct core (punch position 5). Uniquely, microglia are largely unaffected and have comparable KLF9 TF-binding motif enrichment at the infarct core in comparison to the contralateral hemisphere. In addition to KLF9, we also identified STAT3 as varying significantly over space (Supplementary Fig. 5h), which was also an enriched GO term in stroke cells (Fig. 7e). STAT3 has been extensively studied in the JAK/STAT3 pathway, which is a key regulator of apoptosis in cerebral ischemia injuries with reperfusion[50], as well as an initiator of reactive astrogliosis under diverse conditions[51]. Accordingly, we found that STAT3 was largely absent from astrocytes in punches positions 5–7, but was enriched in the reactive astrocytes at the infarct border zone at punch position 8. In contrast, we find that RE1-silencing factor (REST) is significantly elevated at the ischemic core and decreases with proximity to the infarct border. Accordingly, REST has been shown to form a histone deacetylase complex that is a director repressor of SP1 in cerebral ischemia, a TF we identify as varying significantly over space, in the opposite direction of REST[37] (Supplementary Fig. 5i).

Finally, we sought to characterize chromatin accessibility profiles of cells isolated from a single punch at the glial scar (Fig. 8d). To do this, we isolated two punches (punch 40 and punch 48), both originating from the same section (15.SB2), from punch position 8 of the stroke (punch 40) and contralateral hemisphere (punch 48). We processed the cells in isolation as described in prior individual punch analyses, using the peak set from the full experiment. We performed DNA-binding motif enrichment analysis across all cells[52] and then performed cell-type-specific comparisons for a glial (oligodendrocyte) and neuronal (glutamatergic neuron) cell type. In oligodendrocytes, 56 TF motifs were significantly different between the stroke and contralateral hemisphere, many of which (44; 78.6%) corresponded to higher enrichment in stroke as compared to contralateral. Specifically, we found BCL11B (CTIP2), a negative regulator of glial progenitor cell differentiation to be significantly increased at the glial scar (Fig. 8d, left)[53]. Conversely, we found RXRG, a positive regulator of OPC differentiation, and remyelination, to be significantly depleted (Fig. 8d, left)[54]. Together these findings indicate impaired ability of OPCs to differentiate into mature oligodendrocytes at the glial scar. In glutamatergic neurons, we found neuron-associated TFs such as NEUROD2 to be significantly depleted in the stroke hemisphere, which corresponds with decreased neuronal cell types at punch position 8 in the stroke hemisphere. In accordance with our infarct core-to-border axis (punch positions 5–8) analysis, we found that seven of the KLF family of TFs (KLF2–4, 6–8, and 12) were significantly depleted in glutamatergic neurons at the glial scar in the stroke hemisphere (Fig. 8d, right; KLF4 and KLF7 shown). Interestingly, previous studies have found that in response to cerebral ischemia, KLF4, 5, and 6 are induced in astrocytes, while KLF2 is depleted in endothelia and induced in microglia[55]. With these data, we identify that motif enrichment for many members of the KLF family not only significantly vary over space across all cell types, we also indicate novel depletion of multiple KLFs specifically in glutamatergic neurons at the ischemic glial scar.

## Discussion

sciMAP-ATAC provides a low-cost, highly scalable, hypothesis-independent approach to acquiring spatially resolved epigenomic single-cell data with the use of immediately available commercial tools. In addition, sciMAP-ATAC is translatable to any tissue, culture, or model system compatible with cryosectioning. While many methods rely on signal-to-noise optical detection of densely packed molecules and computationally intensive spatial reconstruction, sciMAP-ATAC encodes nuclear localization directly into each library molecule, allowing for rapid subsetting of cells by localization and mapping of cells across vector space in 3D between adjacent sections. We demonstrate the use of sciMAP-ATAC to profile the murine somatosensory cortex, as well as multi-punch trajectories through the human primary visual cortex, recapitulating known marker gene progression through cortical layering, and cell type composition based on the category and positioning of spatially registered microbiopsy punches. We further show the utility of sciMAP-ATAC to resolve the progressive epigenomic changes in a cerebral ischemia model system, revealing distinct trends in chromatin accessibility, cell type composition, and cell states along the axes of tissue damage and altered morphology. Application of sciMAP-ATAC to other highly structured systems or tissues with a gradient of disease phenotype will be particularly valuable areas for this technology. The primary limitation of sciMAP-ATAC is that punches are currently performed manually and registered with adjacent imaged sections post-punching. This limits the precision of desired punch positions, as well as throughput; however, automated processing of tissue sections using robotics[56], where punch patterns are designed on adjacent imaged sections and registered to the target section will enable high precision, as well as increased throughput into the range of thousands. Furthermore, as spatial transcriptomic technologies evolve, they may enable the acquisition of chromatin accessibility information; however, substantial technical hurdles must first be overcome, and profiles produced would be in aggregate over the feature size and not necessarily single cell. Finally, here we applied the sciMAP strategy to assess chromatin accessibility; however, it can, in theory, be applied to any single-cell combinatorial indexing technique to enable spatially registered single-cell genome[19], transcriptome[57], chromatin folding[58], methylation[59], or multi-omic[60–62] assays.

## Methods

**Mouse brain and human VISp cortex sample preparation**. A step-by-step protocol describing the sciMAP-ATAC methods can be found at Protocols.io. All animal studies were approved by the Oregon Health and Science University Institutional Animal Care and Use Committee. Male C57Bl/6 J mice aged 8 weeks were purchased from Jackson Laboratories for the mouse whole brain sciATAC, punch dissociation development sciMAP-ATAC, and mouse SSp cortex sciMAP-ATAC experiments. All mouse cages were kept on a 12 h light/dark cycle at a temperature of 70 °F and within a humidity range of 30–70%. Animals were sacrificed by carbon dioxide primary euthanasia and cervical dislocation secondary euthanasia. Animals were immediately decapitated, intact brain tissue was harvested, washed in ice-cold phosphate-buffered saline (PBS; pH 7.4), submerged in TFM (Cat. TFM-C) within a disposable embedding mold (Cat. EMS 70183). Human VISp cortex samples were provided by the Oregon Brain Bank 5.5 h postmortem and were submerged in TFM. The use of human samples in this study falls under the NIH defined Exempt Human Subjects Research, under Exemption 4 (https://humansubjects.nih.gov/). Embedded mouse whole brain and human VISp

cortex samples were flash-frozen in liquid nitrogen cooled isopentane by lowering the sample into the isopentane bath without submerging within 5 min of embedding. Samples were immediately transferred to dry ice, paraffin wrapped to delay sample dehydration, and stored in an airtight container at −80 °C.

**Mouse cerebral ischemia model**. Two C57BL/6 9-week-old (P63) female mice were placed under isoflurane anesthesia (5% induction, 1.5% maintenance) in 30% oxygen-enriched air. Body temperature was maintained at 37 ± 0.5 °C throughout the procedure. Middle cerebral artery (MCA) occlusion was performed using a previously described method by Longa et al. with slight modifications[63]. Briefly, a laser Doppler flowmeter (Moore Instruments) probe was affixed over the right parietal bone overlying the MCA territory to monitor changes in cerebral blood flow. A midline incision was made, the right common carotid artery (RCCA) bifurcation was exposed by gentle dissection, and the external carotid artery (ECA) was permanently ligated distal to the occipital artery using electrocautery, such that a short ECA stump remained attached to the bifurcation. The RCCA and internal carotid arteries (ICA) were temporarily closed with reversible slip knots before an arteriotomy was made in the ECA stump. A silicone-coated 6.0 nylon monofilament was inserted into the ICA via the arteriotomy and gently advanced to the ICA/MCA bifurcation to occlude CBF to the MCA territory, and confirmed by a laser Doppler signal drop of <30% of baseline. After 60 min occlusion, the filament was gently retracted, the ECA permanently ligated, the slip knot of the common carotid artery removed, and the incision sites sutured closed. The mice exposed to MCAO were euthanized 3 days after the MCAO procedure, intact brain tissue harvested, washed in ice-cold PBS (pH 7.4), submerged in TFM, and flash-frozen in liquid nitrogen cooled isopentane. Samples were paraffin wrapped and stored at −80 °C and intact embedded whole mouse brains were sectioned at the time of experiment.

**Sample sectioning**. All embedded samples were sectioned in a cryostat (Leica CM3050) at −11 °C chuck and chamber temperature and collected on Superfrost Plus microscope slides (Fisherbrand, Cat. 22-037-246). Sectioning was performed in sets of: one section at 100–300 µm paired with three sections at 20 µm, to generate sets of four slides consisting of microbiopsy (1) and histology (3) sections at one section per slide. Slide boxes were sealed with paraffin to prevent sample dehydration and stored long term at −80 °C.

**Mouse whole brain coronal section immunohistochemistry and mapping**. To determine the mouse brain atlas coordinate of each coronal microbiopsy section, the histological section immediately adjacent to each microbiopsy section were fixed in 4% PFA for 10 min and counterstained using 300 µM DAPI (Thermo Fisher, Cat. D1306) in 1× (pH 7.4) PBS (Thermo Fisher, Cat. 10010023) for 5 min. Slides were rinsed with 1× PBS and mounted in Fluoromount-G (Thermo Fisher, Cat. 00-4958-02). Slides stained for Satb2 were equilibrated to room temperature and circumscribed with a hydrophobic barrier pen (Invignome, Cat. GPF-VPSA-V). Sections were washed twice with PBS for 10 min then blocked for 1 h at room temperature in permeabilization/blocking buffer comprised of PBS with 10% normal goat serum (NGS, Jackson ImmunoResearch, Cat. 005-000-121), 1% bovine serum albumin (BSA, Millipore, Cat. 126626), 0.3% Triton X-100 (TX-100, Sigma, Cat. 11332481001), 0.05% Tween-20 (Sigma, Cat. P1379), 0.3 M glycine (Sigma, Cat. G7126), and 0.01% sodium azide (Sigma, Cat. S2002). During the blocking step, the primary antibody rabbit anti-Satb2 (Abcam Cat. ab92446) was diluted 1:1000 in a buffer containing PBS, 2% NGS, 1% BSA, 0.01% TX-100, 0.05% Tween-20, and 0.01% sodium azide. The diluted primary antibody was applied to sections then incubated overnight at 4 °C. The primary antibody was washed from the sections five times with PBS for 5 min at room temperature. Secondary antibody AF488 goat anti-rabbit (Thermo Fisher Cat. A32731) was prepared by diluting 1:1000 in the same buffer used to dilute primary antibodies. Sections were incubated with the diluted secondary antibody for 1 h in the dark at room temperature. Secondary antibodies were washed from the sections three times with PBS for 5 min, then nuclei were counterstained with DAPI for 10 min at room temperature. After DAPI staining, sections were washed an additional two times then glass coverslips were mounted with ProLong Diamond Anti-Fade Mounting Medium (Thermo Fisher, Cat. P36961). Slides were imaged on a Zeiss ApoTome AxioImager M2 fluorescent upright microscope and processed using Fiji software (v1.52p)[64]. Coronal section images were mapped to the Adult Mouse Allen Brain Atlas[25] according to anatomical regions.

**Mouse cerebral ischemia immunohistochemistry and mapping**. One of the histological sections corresponding to each microbiopsy section was stained for GFAP to identify the infarct. Slides were equilibrated to room temperature and circumscribed with a hydrophobic barrier pen. Sections were washed twice with PBS for 10 min then blocked for 1 h at room temperature in permeabilization/blocking buffer comprised of PBS with 10% normal donkey serum, 1% BSA, and 0.05% TX-100. The sections were next incubated in primary antibody solution comprised of 1:1000 goat anti-GFAP (Abcam, ab53554) and 1:5000 rabbit anti-Iba1 (Fujifilm Wako, NCNP24) diluted in PBS with 1% NGS, 0.1% BSA, and 0.005% TX-100 overnight at 4 °C. The sections were then washed three times with PBS for 5 min each at room temperature and next incubated for 2 h at room

temperature in secondary antibody solution containing 1:500 donkey anti-goat conjugated to Alexa Fluor 488 (Invitrogen) and 1:500 donkey anti-rabbit conjugated to Alexa Fluor 555 (Invitrogen) prepared in the same buffer as the primary antibodies. Following the secondary incubation, sections were washed three times with PBS for 5 min each, counterstained with DAPI for 10 min, washed an additional two times with PBS for 5 min each, then coverslipped with Fluoromount-G. Slides were imaged on a Zeiss AxioScan.Z1 Slide Scanner and processed using Fiji software (v1.52p). Coronal cerebral ischemia section images were mapped to the Adult Mouse Allen Brain Atlas[25] according to anatomical regions using the DAPI channel, as described above.

Immunohistochemistry fluorescence was quantified using Fiji software (v1.52p). Punch positions were mapped to regions of interest (ROIs), along with three negative naive ROIs for each image. Corrected total fluorescence was calculated as the difference between the integrated density (ROI area × mean fluorescence) of an ROI for a given punch and the average integrated density of negative naive ROIs. GFAP-corrected total fluorescence was plotted using *geom_boxplot* and *geom_smooth*, method *lm* using *ggplot* (v3.2.1) in R (v3.5.1).

**Mouse whole brain dissociation and nuclei isolation**. To evaluate the effect of flash-freezing on chromatin accessibility in mouse brain tissue, we evaluated single-cell chromatin accessibility profiles from an intact mouse brain, in which one hemisphere was flash-frozen and one hemisphere remained unfrozen. Both hemispheres were processed in parallel and underwent dissociation and nuclear isolation. Tissue was diced in NIB (10 mM Tris HCl, pH 7.5 [Fisher, Cat. T1503 and Fisher, Cat. A144], 10 mM NaCl [Fisher, Cat. M-11624], 3 mM MgCl$_2$ [Sigma, Cat. M8226], 0.1% IGEPAL [v/v; Sigma, I8896], 0.1% Tween-20 [v/v, Sigma, Cat. P7949], and 1× protease inhibitor [Roche, Cat. 11873580001]) in a petri dish on ice using a chilled razor blade. Diced tissue was transferred to 2 mL chilled NIB in a 7 mL Dounce-homogenizer on ice. The tissue was incubated on ice for 5 min then homogenized via 10 gentle strokes of the loose pestle (A) on ice, a 5-min incubation on ice, then ten gentle strokes of the tight pestle (B) on ice. The homogenate was then strained through a 35 µm strainer and centrifuged at 500 × g for 10 min. Samples were aspirated, resuspended in 5 mL of ice-cold NIB, and nuclei were counted on a hemocytometer. Samples were diluted to 500 nuclei per 1 µL to facilitate tagmentation reaction assembly at ~5000 nuclei per 10 µL of NIB.

**Tissue microbiopsy acquisition and nuclear isolation**. Tissue microbiopsies were acquired from 100–300 µm sections. Punches were isolated in four experiments: (1) mouse dissociation development sciMAP-ATAC (384 punches), (2) mouse SSp cortex sciMAP-ATAC (96 punches), (3) mouse cerebral ischemia sciMAP-ATAC (240 punches), and (4) human VISp cortex sciMAP-ATAC (192 punches; for details refer to Supplementary Fig. 1). Microbiopsy coronal sections were acclimated to −20 °C in a cryostat (Leica CM3050) and microbiopsy punch tools (EMS, Cat. 57401) were cooled on dry ice prior to punching to prevent warming of tissue. Microbiopsy punches were acquired according to location identified from section atlas mapping, and frozen microbiopsies were deposited directly into 100 µL of ice-cold NIB in a 96-well plate. Punch deposition into each well of the 96-well plate was visually confirmed under a dissecting microscope. To facilitate tissue dissociation and nuclear isolation, 96-well plates of microbiopsy punches were then gently shaken (80 r.p.m.), while covered for 1 h on ice. We then tested mechanical dissociation by varying the number of triturations performed via multichannel pipette per well (punch dissociation development sciMAP-ATAC). We found the following averaged metrics across the four dissociation methods: 15 triturations (26 cells per punch, 5679 unique passing reads per cell, 0.844 FRis), 30 triturations (35 cells per punch, 7189 unique passing reads per cell, 0.835 FRis), 60 triturations (28 cells per punch, 7611 unique passing reads per cell, 0.827 FRis), and 100 triturations (8 cells per punch, 7611 unique passing reads per cell, 0.821 FRis). Given that 60 trituration mechanical dissociation yielded the highest number of cells per punch, with otherwise comparable metrics, we proceeded with 60 triturations for all future experiments. Post-mechanical dissociation, sample plates were then centrifuged at 500 × g for 10 min. While nuclear pellets were not visible, we found that aspiration of 90 µL of supernatant and resuspension in an added 30 µL of NIB results in a final isolated nuclear volume of 40 µL with ~15,000 nuclei per well (for microbiopsy punching conditions: 200 µm section, 250 µm diameter microbiopsy punch used in the human VISp and mouse cerebral ischemia preparations). Nuclei were split across four 96-well plates such that nuclei were aliquoted to 10 µL, or ~3750 nuclei per well. This enabled four independent indexed transposase complexes to be utilized for each individual punch, or 384 uniquely indexed transposition reactions in one experiment. To calculate the approximate resolution for each preparation, we took the cubed root of the cylindrical volume.

**Location indexing via tagmentation**. Transposase catalyzed excision of the chromatin accessible regions via tagmentation results in the addition of unique molecular identifiers (indexes) for each tagmentation reaction. Uniquely indexed transposase adapter sequences are reported in Supplementary Table 2. To encode microbiopsy punch location into library molecules, we recorded the corresponding tagmentation well within each 96-well plate to the user-identified microbiopsy punch location. The incorporation of location information is therefore inherently encoded by the first tier of indexing in our established sci-ATAC-seq method.

Tagmentation reactions were assembled at 10 µL of isolated nuclei at 500 nuclei per 1 µL, 10 µL 2× tagmentation buffer (Illumina, Cat. FC-121-1031), and 1 µL of 8 µM loaded indexed synthesized Tn5 transposase was added per well (See Picelli et al. for transposase synthesis protocol)[65]. As an alternative to Tn5 synthesis, EZTn5 transposase (https://www.lucigen.com/EZ-Tn5-Transposase/) can be purchased commercially and diluted, salt adjusted, and loaded with sci indexes according to the sciMAP-ATAC protocol[66]. Each assembled 96-well plate of tagmentation reactions was incubated at 55 °C for 15 min. For the mouse whole brain sci-ATAC-seq preparation on fresh and frozen tissue, as well as the sciMAP-ATAC preparations, four 96-well plates of tagmentation were used (384 uniquely indexed tagmentation reactions). For whole brain sci-ATAC-seq preparation on fresh and frozen tissue experiment, tagmentation wells were pooled separately for fresh and frozen hemisphere samples. For the microbiopsy punch-derived experiments, all reactions were pooled post-tagmentation.

**Combinatorial indexing**. To lyse nuclei and release bound transposase, PCR plates are prepared with protease buffer (PB), primers, and sparsely sorted nuclei and then incubated. Uniquely indexed PCR primer sequences are reported in Supplementary Table 2. Post-denaturation, the remaining PCR reagents are added and incorporation of the PCR primers results in incorporation of the secondary index for single-combinatorial indexing. For the denaturation step, 96-well PCR plates of 8.5 µL PB (30 mM Tris HCl, pH 7.5, 2 mM EDTA [Ambion, Cat. AM9261], 20 mM KCl [Fisher, Cat. P217 and Fisher, Cat. A144], 0.2% TX-100 [v/v], 500 µg/mL serine protease [Fisher, Cat. NC9221823]), 1 µL 10 mM indexed i5, and 1 µL indexed i7 per well were prepared. Pooled tagmented nuclei were stained by adding 3 µL of DAPI (5 mg/mL) per 1 mL of sample. Each sample was then FAN sorted using BD FACSDiva software (v8.0.1) on a Sony SH800 FACS machine at 22 events per well per 96-well Tn5 plate (e.g., 88 for 384 indexes) into prepared 96-well plate(s). Event numbers were selected based on the expected success rate of events as actual cells for a given target cell doublet rate (see "Doublet rate estimations" section below). Across the sciMAP-ATAC experiments, four PCR plates (384 uniquely indexed wells) were utilized for the initial punch-derived sci-ATAC-seq preparation from whole brain-derived punches, two PCR plates (192 uniquely indexed wells) were used for the mouse SSp cortex experiment, one full and one partial plate (128 uniquely indexed wells) for the human VISp experiment, two plates (192 uniquely indexed wells) for the mouse cerebral ischemia experiment, and finally two PCR plates (192 uniquely indexed wells) were utilized for the nonspatial whole brain sci-ATAC-seq preparation on fresh and frozen tissue. Transposase denaturation was performed by sealing each sorted plate and incubating at 55 °C for 15 min. Plates were immediately transferred to ice post-incubation and 12 µL of PCR mix (7.5 µL NPM [Illumina Inc. Cat FC-131-1096], 4 µL nuclease-free water, and 0.5 µL 100× SYBR Green) was added to each well. For each experiment, plates were then sealed and PCR amplified on a BioRad CFX real-time cycler running CFX Manager (v3.1) software, using the following protocol: 72 °C for 5:00, 98 °C for 0:30, cycles of (98 °C for 0:10, 63 °C for 0:30, 72 °C for 1:00, plate read, 72 °C for 0:10) for 18–22 cycles. PCR plates were transferred to 4 °C once all wells reached mid-exponential amplification on average. Each PCR plate is then pooled at 10 µL per well and DNA libraries are isolated using a QIAquick PCR Purification column. Each pooled PCR plate library is then quantified using a Qubit 2.0 fluorimeter, diluted to 4 ng/µL with nuclease-free water, and quantification of library size performed on an Agilent Bioanalyzer using a dsDNA high sensitivity chip. Libraries were then sequenced on a NextSeq^TM 500 sequencer (Illumina Inc.) running NextSeq500 NCS (v4.0) software loaded within a range of 1.2–1.6 pM with a custom sequencing chemistry protocol (read 1:50 imaged cycles; index read 1:8 imaged cycles, 27 dark cycles, 10 imaged cycles; index read 2:8 imaged cycles, 21 dark cycles, 10 imaged cycles; read 2:50 imaged cycles) using custom sequencing primers supplied in Supplementary Table 2.

**Doublet rate estimations**. An important factor in single-cell studies is the expected doublet or collision rate. This manifests in droplet-based platforms as two cells being encapsulated within the same droplet, thus having the same cell barcode for their genomic information. This is tunable by the number of cells or nuclei loaded onto the instrument, with typical doublet rates targeted to be at or <5%. This is also true for combinatorial indexing workflows, where doublets are present in the form of two cells or nuclei with the same level 1 index—which is the transposase index for ATAC—that end up in the same level 2 indexing well (i.e., the PCR well). This results in an identical pair of indexes for the two cells. This rate, like with droplet methods, is also tunable by altering the number of indexed cells or nuclei that are deposited into each well, with a typical experiment targeting at or below a 5% doublet rate. This rate is approximated by leveraging the "birthday problem" formulation in statistics, where the transposase index space (days in the year) and number of indexed nuclei per well (number of people at each table) are taken into account. These predictions assume that there is complete mixing of nuclei prior to distribution and that the distribution is unbiased, which is reasonable given the single-nuclei suspension and use of flow sorting for the distribution process, and hold up when compared to empirical data produced by multispecies cell mixing experiments[18,19,59] (i.e., barnyard experiments, typically mixing human and mouse cells). However, in the case of sciMAP-ATAC, nuclei are directly isolated and then indexed within the same well, making a true barnyard experiment not feasible. Any experiment that would use tissue punches from two

different species into different wells would not capture doublets because of the de facto unique indexes for each species imparted by the different wells for the first level of indexing. We therefore assumed that the assumptions that have been made and tested for standard sci-ATAC-seq and related combinatorial technologies also apply to sciMAP-ATAC, as the novel components of the workflow are in the processing prior to the combinatorial indexing stages.

With our set of 384 unique transposase indexes and the sorting of 88 nuclei per well across experiments, this would result in a doublet rate (i.e., two nuclei of the same transposase index ending up in the same PCR well) of 10.5% if the yield of sorted nuclei was perfect. However, we favor speed over precise quantification during the sorting step, as the actual number of sorted cells does not matter as long as it ends up being below the target number. We have found that using our fast-sorting workflow, of the target number of events that are sorted, only between 25 and 50% are true nuclei. The rest of the events are empty droplets. We also note that these droplets do not contain ambient chromatin based on human–mouse mixing experiments[19]. Using the high end of the ~50% true nuclei yield, the expected doublet rate is 5.4%, in line with other commercially available single-cell platforms. When factoring in the actual yield with respect to single-cell profiles produced, the doublet rate is even lower. For example, the punch dissociation development sciMAP-ATAC preparation produced 8012 single-cell profiles over 384 unique indexed transposition wells, for an average of just under 21 cells produced per well out of the 88 events that were sorted—a 23.7% yield. The final expected doublet rate is therefore most accurately calculated according to 21 indexed nuclei produced per well with a transposase index space of 384 for a doublet rate of 2.5%, which is well within the accepted range.

**Sequence data processing**. Fastq files were generated from BCL files using bcl2fastq (Illumina Inc., v2.19.0). Fastq files were aligned, filtered, and analyzed primarily using the "scitools" software (github.com/adeylab/scitools)[22], which includes wrappers for numerous external tools. Raw sequence reads had their index combinations matched to a whitelist of expected indexed using "scitools fastq-dump", which allows for a hamming distance of two and produces error-corrected fastq files. These were then aligned to a mouse or human reference genome (mm10 or hg38) via bwa mem (v0.7.15-r1140)[67] and sorted using "scitools align". PCR duplicate removal and filtering for quality ten aligned autosomal and chromosome X reads (i.e., excluding mitochondrial, chromosome Y, and unanchored contigs) was performed using "scitools rmdup" with default parameters and plotted using "scitools plot-complexity". Projections of passing reads given increased sequencing depth was performed using "scitools bam-project" on the pre-duplicate removed bam file, which generates a model for every single cell based on sampling reads and calculating the passing read percentage that empirically falls within 2% accuracy[19]. Bam files were then filtered to only contain cell barcodes that contained a minimum of 1000 passing reads and a percent unique reads <80 (any overly complex cell libraries may be doublets and were therefore excluded). For the human VISp dataset, cells were also filtered to have a TSS enrichment (per cell calculation) of 2 (see section "Quality metric calculations" below).

**Chromatin accessibility analysis**. The filtered bam file was used for chromatin accessibility peak calling for each of the five experiments individually, as well as on a combined bam file from the mouse whole brain sciATAC-seq, mouse punch dissociation development sciMAP-ATAC, and mouse SSp cortex sciMAP-ATAC experiments for the combined dataset analysis. Peak calling was run using the wrapper function "scitools callpeak", which utilized macs2 (v2.1.1.20160309) for peak calling, and then filtering and peak extension to 500 bp (ref. [68]). Called peaks from mouse whole brain sciATAC-seq, mouse punch dissociation development sciMAP-ATAC, and mouse SSp cortex sciMAP-ATAC datasets were merged to generate a union peak set that was used to compare sciATAC-seq and sciMAP-ATAC clustering. Peak bed files and filtered bam files were then used to construct counts matrix of cells × peaks. Latent Dirichlet Allocation using the package cis-Topic (v0.2.0)[69] was performed using the scitools wrapper function "scitools cis-topic". Topic enrichments for region type annotations (Supplementary Fig. 4g) were annotated using cisTopic function *annotateRegions*, using the Bioconductor package *TxDb.Hsapiens.UCSC.hg38.knownGene* (v3.4.7) and annotation database *org.Mm.eg.db* (v3.8.2). The topic by annotation heatmap was plotted using cisTopic function *signaturesHeatmap*. The cells × topics matrix was biclustered and plotted using "scitools matrix-bicluster", which utilizes the *Heatmap* function in the *ComplexHeatmap* package (v1.20.1) in R (v3.5.1)[70]. Two-dimensional visualization was performed using UMAP via "scitools umap" and plotted using "scitools plot-dims". Visualization of topic weights on the UMAP coordinates was performed using "scitools plot-dims" with -M as the cells × topics matrix. Clustering was performed on the cells × topics matrix using the package *Rphenograph* (v0.99.1) in R (v3.5.1), which employs Louvain clustering and was executed using the wrapper function "scitools matrix-pg"[71]. In addition to topic analyses, we utilized ChromVAR (v1.4.1)[52] to assess the global motif accessibility profiles of cells using the wrapper function "scitools chromvar" on the bam file with added read group tags using "scitools addrg". Boxplots illustrating TFME per cell were generated using values from the ChromVAR deviations_scores matrix and plotted using *geom_boxplot* from the package *ggplot* (v3.2.1) in R (3.5.1), where lower and upper hinges indicate first and third quartiles, center line indicates median, upper, and lower whiskers indicate 1.5 times the inner quartile range (IQR). Data points

beyond the end of the whiskers are plotted individually. All boxplot comparison significance calculations were performed using the *compare_means* function in the *ggpubr* package (v0.2.5) indicating *paired = FALSE* and *p.adjust.method* set to Bonferroni–Holm correction in R (v3.5.1).

**Quality metric calculations**. To generate tagmentation site density plots centered around TSSs, we first subset filtered experiment bam files into respective annotations. We used the alignment position (chromosome and start site) for each read to generate a bed file that was then fed into the BEDOPS closest-feature command mapped the distance between all read start sites and TSSs (v2.4.36, ref. [72]). From this, we collapsed distances into a counts table respective of experiment and annotation, and generated percentage of read start site distances within each counts table. We plotted these data using R (v3.6.1) and *ggplot2* (v3.3.2) *geom_line* function (default parameters) subset to 2000 base pairs around the start site to visualize enrichment. TSS enrichment values were calculated for each experimental condition using the method established by the ENCODE project (https://www.encodeproject.org/data-standards/terms/enrichment), whereby the aggregate distribution of reads ±1000 bp centered on the set of TSSs is then used to generate 100 bp windows at the flanks of the distribution as the background and then through the distribution, where the maximum window centered on the TSS is used to calculate the fold enrichment over the outer flanking windows. The fraction of reads in a defined read set (FRiS) was used as an alternative to the fraction of reads in peaks for two major reasons. The first is that FRiP is highly dependent on the number of peaks that are called, which is, in turn, highly dependent on (a) the number of cells profiled, and (b) the depth of sequencing. One can increase FRiP values by sequencing a library more deeply or profiling larger numbers of cells at the same depth without reflecting any difference in underlying data quality. Second, peak calling on a population of cells favors peaks in high abundance cell types, as they make up more of the data going into the peak calling. Therefore, cells of a cell type that is lower abundance will have fewer peaks called that are specifically associated with that cell type, owing to the dominance of signal by the more abundant cell type and consequently reducing the FRiP of those cells. Using FRiS instead largely avoids the challenges associated with peak calling by leveraging a comprehensive reference dataset. For the mouse FRiS calculations, we aggregated peaks that are available from mouse bulk ATAC-seq and DNAse hypersensitivity experiments provided by the ENCODE project, followed by peak collapsing, resulting in 2,377,227 total peaks averaging 744.9 bp. For the human dataset, we used a human reference dataset for DNAse hypersensitivity[28] that contains 3,591,898 loci defined as TF footprints with an average size of 203.9 bp leading to the lower FRiS values when compared to the aggregate mouse ATAC-seq peak dataset.

**Cell type identification**. The identified clusters were assigned to their respective cell type by examining the chromatin accessibility profile of marker genes that correspond to known cell types. Gene regions were plotted using "scitools plot-reads" using the filtered bam file and genome track plots were generated using *CoveragePlots* from the analysis suite of tools, *Signac* (v0.2.5, https://github.com/timoast/signac). Additional support for identified cell types was performed by assessing the chromVAR results for global motif accessibility. Marker genes used for cell type identification included: *Gfap*, *Glul*, and *Agt* for astrocytes, *Col19a1* for all neuronal cell types, *Gad1*, *Gad2*, *Pvalb*, *Dlx1*, and *Dlx2* for GABAergic neurons, Slc17a7, *Drd1*, *Drd2*, *Bcl11b* (*Ctip2*), and *Ppp1r1b* for GABAergic MSNs, also referred to as SPNs, *C1qa*, *C1qc*, and *Cx3cr1* for microglia, *Mrc1* for macrophages within the microglia cluster, *Kdr* and *Flt1* for endothelia, *Olig1* for al oligodendrocyte cell types, *Top2a* and *Cspg4* (NG2) for OPCs, *Fyn*, and *Prox1* for newly formed oligodendrocytes, and *Mobp*, *Mog*, *Cldn11*, and *Prox1* for mature myelinating oligodendrocytes.

**Gene ontology enrichment analysis**. GO enrichment analysis was performed for the genomic regions defined within topic 30, the topic enriched in ischemia specific cells. Single nearest genes to topic 30 regions were identified using GREAT (v4.0.4) for reference genome mm10 (ref. [73]). GO term statistical overrepresentation for GO biological processes was calculated using Panther (v14) binomial test with false discovery rate (FDR) correction for overrepresentation of topic 30 genomic regions in comparison to all mouse (mm10) genes. Data were plotted using *ggplot* (v3.2.1) plotting function *geom_barplot* in R (v3.5.1) with height corresponding to log₂ GO term fold enrichment and colored by GO term $-\log_{10}$ FDR *Q*-value.

**Transcription factor and site enrichment through trajectories**. TF motif enrichment analysis was performed using chromVAR (v1.4.1) in R (v3.5.1) on all cells derived from ischemia mouse models, including the ischemic (stroke) hemisphere and contralateral (contra) hemisphere. For the cells × TFME matrix, cells were annotated by the punch they were derived from, and a linear regression of TFME as a function of punch location for each cell using the base function *lm* in R (v3.6.1). Slopes of the linear model for the ischemic and contralateral hemispheres were defined as the coefficient of the fit. The statistical significance of the interaction between TFME over space and disease condition (stroke versus contralateral hemisphere) was calculated by performing an analysis of variance (ANOVA, *anova* base R v3.6.1) on the interaction of hemisphere on the linear regression defined by

TFME as a function of punch position (TFME ~ punch × hemisphere$_{(stroke/contra)}$), and slopes were compared using the lsmeans package function *lstrends* (v2.30-0). Slopes were compared between the stroke and contralateral hemispheres by taking the difference between the slopes (Δ slope = slope$_{stroke}$ − slope$_{contra}$). The change in slope was *z*-scored to center and scale TFME difference, where *z*-score Δ slope is equal to two standard deviations from the mean. Volcano plot of $-\log_{10}$ *p* value by Δ slope was generated using the package EnhancedVolcano (v1.4.0) in R (v3.5.1). Line plots vignettes were generated by plotting volcano plot data using *ggplot* (v3.2.1) plotting function *geom_smooth*, method *lm*. Heatmaps illustrating cell-type-specific TFME over space were generated by subsetting ischemia mouse model cells by cell type, and plotting TFME by punch, compared between stroke and contralateral hemispheres using package ComplexHeatmap (v2.0.0) in R.

Analysis of putative regulatory elements was performed by assessing the ATAC peak probabilistic weight per cell (*cisTopic* predictive distribution) across cells derived from punches taken from the infarct core to infarct border axis (punch positions 5–8) in the stroke and contralateral hemispheres, aggregated across all MCAO mice. This was performed similarly to TFME described above, where ATAC peak probability per cell was averaged by punch position (punch positions 5–8). ATAC peak probability along the 5–8 axis was fit to a linear model and the slope in the stroke hemisphere was compared to the slope in the contralateral hemisphere in order to generate significance and delta-slope values. We found that 3852 peaks out of 104,773 total peaks (4.8%) vary significantly across the 5–8 axis in MCAO stroke hemispheres in contrast to the contralateral hemispheres. In order to identify putative regulatory elements which are associated with the progressive gradient of glial reactivity from the infarct core to the infarct border in stroke hemispheres, we subset our spatially significant peak set to those which uniformly increase or decrease along the 5–8 axis in stroke hemispheres. We found 73 sites which uniformly increase with increasing proximity to the infarct border, and no sites which uniformly decrease. We report these 73 spatially significant peaks as a reference for future MCAO regulatory element studies.

**Moran's *I* spatial autocorrelation analysis**. We performed a Moran's *I* test to assess spatial autocorrelation between punch locations, wherein a higher Moran's *I* value signifies a higher chance of cells from the same punch location being nearby in Euclidean space. Cells sourced from white matter punches were excluded. The test was performed using the same 27 topic weight matrix used for UMAP projections. Cells were split by assigned types and processed in parallel in R (v4.0) using a modified version of the "Moran_*I*" method in the function graph_test in monocle3 (v0.2.3.0)[74]. Briefly, we used a bootstrapping method wherein each punch location (1–8) was randomly assigned a new location for all punches and all trajectories, such that all cells from the same punch still shared the same location. The Moran's *I* value was calculated for 1000 iterations using this random location reassignment strategy. The resulting null distribution was then compared to our true punch location Moran's *I* using the pnorm function to perform an unpaired one-sided (lower.tail = FALSE) *z*-test. To account for multiple testing, we applied a Bonferroni correction to the *p* values.

**Integration with snDrop-seq and scTHS-seq data**. We applied cross-data-modality integration based in canonical correlation analysis (CCA) to coanchor our sciMAP-ATAC-seq dataset with publicly available snDrop-seq and scTHS-seq visual cortex datasets[29,75] (Fig. 5). For single-cell chromatin accessibility data, we used Signac (v1.1.0)[75] to perform latent semantic indexing (LSI) on the filtered chromatin accessibility matrices (for both scTHS-seq and sciMAP-ATAC-seq) and calculated the normalized LSI loadings scores (using dims: 2:30) for anchor weighting. We then created gene activity matrices using the R package Cicero (v1.3.4.10)[76]. Similarly, we preprocessed the snDrop-seq expression matrix using the standard Seurat 3 (v3.2.1) workflow, where we filtered for variable features (5000 features), scaled and normalized data, reduced dimensions via PCA and UMAP. For RNA-ATAC integration, we first learned the transfer anchors based on the gene activity and expression data by applying FindTransferAnchor (with the parameters dims = 1:30 and reduction = "cca"). We then used TransferData (weight.reduction = atac[["lsi"]]), to project scRNA-seq data labels onto sciMAP-ATAC-seq cells. We finally created a confusion count matrix based on the label matches between the snDrop-seq predicted and sciMAP-ATAC-seq labels. Using a similar method for feature imputation at variable genes, we transferred the scRNA-seq data onto the sciMAP-ATAC-seq cells and performed PCA on the combined datasets, followed by visualization via UMAP[77]. We applied a matching CCA-based strategy to coanchor scTHS-seq and sciMAP-ATAC-seq cells, using 70,832 overlapping accessibility sites between the datasets. For label transfer, we used the normalized LSI loadings scores for anchor weighting of the scTHS-seq data and then compared labels via a confusion matrix (Fig. 5).

**Statistics**. Data are presented as mean ± SEM unless otherwise specified. Statistical analysis was generally performed by two-sided, unpaired Wilcoxon nonparametric test or two-way ANOVA and the Bonferroni method of correction for pairwise multiple comparisons, or as specified in the figure legends. Significance was assigned to $p < 0.05$. All analyses were performed using R version 3.5.1 or scitools scripts (github.com/adeylab/scitools) unless otherwise specified. Plots were generated primarily using R ggplot2 version 3.2.1.

**Reporting summary**. Further information on research design is available in the Nature Research Reporting Summary linked to this article.

## Data availability

Raw and processed single-cell library sequencing data, as well as single-cell metadata have been submitted to the National Center for Biotechnology Information Gene Expression Omnibus (GEO) under the accession code GSE164849. All other data supporting the findings of this study are available with the article and its Supplementary Information files, and from the corresponding author upon reasonable request. Source data are provided with this paper.

## Code availability

Data analysis and plotting was performed using functions contained within the publicly available scitools software suite of single-cell analysis tools (github.com/adeylab/scitools).

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

## Acknowledgements

We thank members of the Adey, O'Roak, and Wright labs for their support; Kylee Rosette for assistance with animal husbandry; Eleonora Juarez for discussion on protocol development; Anthony P. Barnes and Brian J. O'Roak for helpful discussions on experimental design; and Dominica Cao and Brooke DeRosa for IHC staining. This work was supported by the NIH Brain Initiative, National Institute for Drug Abuse (1R01DA047237), and the NIH National Institute for General Medical Studies (R35GM124704) to A.C.A.; and an OHSU Early Independence Fellowship to C.A.T.

## Author contributions

A.C.A. and C.A.T. conceived of the idea. C.A.T. performed all experiments described with assistance from R.M.M. and A.J.F.; F.J.S., K.M.W. and A.M. contributed to experimental design and data interpretation. C.A.T. performed data processing and analysis with assistance from K.A.T., R.M.M., A.N. and E.G.L. W.Z. and H.M. performed stroke surgeries. R.W. identified, isolated, and cryopreserved human primary visual cortex with assistance from C.A.T. C.A.T. and A.C.A. wrote the manuscript with input from all authors.

## Competing interests

F.J.S. is an employee of Illumina Inc. All other authors declare no competing interests.
