## [Peer Review File · Nature Communications]

Editorial Note: This manuscript has been previously reviewed at another journal that is not operating a transparent peer review scheme. This document only contains reviewer comments and rebuttal letters for versions considered at *Nature Communications*. Mentions of prior referee report have been redacted.

Reviewers' Comments:

Reviewer #2:

Remarks to the Author:

Manuscript by Thornton et al has now been revised, and the authors have added substantial amount of new data, including from primary human visual cortex. Given this substantial change and divergence from the original flow of the manuscript, I am concerned that the manuscript has drifted quite a bit away from the original purpose.

Continuing to evaluate this manuscript as a methods paper, I am not convinced that the approach presented represents a new method. Taking punch biopsies out of the tissue followed by single cell omics hardly constitutes a new method. Incorporation of covalent barcodes has been utilized extensively by the Shendure lab, and again is not in itself particularly innovative.

In the revised manuscript, there are now three datasets presented, mouse somatosensory cortex, human visual cortex, and mouse ischemia model. It is unclear why the authors decided to focus on these three specifically, and what unique advantages they bring. Evaluating this manuscript as a resource paper, there are several valuable resources generated that would be beneficial to the research community, but are relatively underdeveloped. For example, given that the authors now include datasets from the cerebral cortex of two species, I would expect the authors to perform some form of comparative analysis to comment about the spatial organization, or epigenomic similarity across species. This to my knowledge is the first paper that would perform such analysis.

In the initial tissue biopsy optimization, the authors generate a dataset consisting of 8,011 cells sequenced at 12,052 reads per cell. The authors note that this compares well to the sci-ATAC-seq library, which the method is originally based on. This suggests that the method of nuclei isolation upstream of nuclei processing does not appear to negatively impact data quality. They also compare four dissociation methods, and note that the methods can vary substantially - it would be helpful to the readers to comment on how their nuclei isolation method compares to the recent paper from Slyper et al (Nature Medicine PMID: 32587393). Given the updated details regarding biopsy preparation, I am curious what percentage of cells/nuclei are lost in the processing - on page 4 the authors indicate that a mean of 112 cells pass quality control metrics per punch.

There are also several concerns regarding the new analysis introduced in this paper, which is the human brain tissue. The source of the tissue is the Oregon Brain Bank, and they indicate that the PMI was 5.5 hours, and that the sample came from a 60 yr. male. Due to the lack of a biological replicate (N=1), it is impossible to assess the reproducibility of their findings. In order to overcome this clear limitation, it will be important to leverage previously published datasets, such as the one generated by Kun Zhang's lab (Lake et al. 2018 PMID: 29227469). The analysis of the data is remarkably superficial. The authors note that FOS motif is enriched in upper layers, NEUROD6 is enriched in lower layer, and RORB is enriched in 'middle punches'. I am not aware of any evidence that FOS expression is enriched in upper layers of the cerebral cortex, and the fact that this is an immediate early gene raises the possibility of uneven tissue degradation or some other technical artifact. As the authors mention in their description of ischemic mouse data, FOS expression may indicate neuronal damage.

Another major concern in this analysis is the profound depletion of astrocytes. Astrocytes to neuron ratio in the human visual cortex (please indicate specifically what area was profiled) should be approximately 0.6 to 1 depending on what area was profiled. The authors should evaluate the estimates of abundance of these major cell types using adjacent tissue samples by immunostaining. This should not be difficult given that they only annotate major cell classes, not cell types (please note that for clarity).

Taken together, the human cortex dataset featured in this revised manuscript falls short of making a compelling case for their method, and the lack of biological replicates raises the question about the quality of the data obtained from this post mortem sample.

Reviewer #3:

Remarks to the Author:

The revision has substantially improved the original manuscript in: 1, quality control for the sciMAP-ATAC sequencing data, 2, punch-wise analysis results, 3, figure styles and colors, as well as reference to figures in the main text. But still some problems remain.

1, Extended figure 2. quality control. Please also show fragment length distribution. Do you observe a pattern of "nucleosome wave"? This is often used as indicative of ATAC-seq library quality.

2, the analyses in figure 2j and 2k are punch-wise, which is very nice. But these are just two examples. Is it a way to invent a systematic analysis that covers all punches? More importantly, how do you know that these differences between punches are real? How to relate these findings to biology? What do these analyses predict?

3, continue with the above, generally the punch-wise analyses are still limited, and often descriptive. Is it possible to some systematic analyses that really use the information from individual punches?

4, figure 3a-c, still, the results lack of visualization and discussion in punch-level. At least, is it possible that the authors focus on the outmost punches of all trajectory and see how punches of the same layer but different positions vary?

5, still figure 3b, it is interesting to see different punch orders mix randomly in the cluster Olig (define this!) but display a pattern in the cluster Glut. Is there any biology?

6, the paper is too long. It needs to be more concise. Consider to split figures (not to put everything into one figure, which is tiring when reading), also, similar reason, consider to split some ultra-long sections.

Minor:

1, line 188-190, use the same name or define acronyms in the main text and figure legends. Otherwise, who knows the symbols in the figure stand for? This happens to many panels. This seems to be a general problem.

2, Figure 3f, n=168?

3, Lines 291-306, this introductory paragraph is too long.

General Response:

We thank the reviewers for their additional comments and suggestions. Major updates include additional visualization and analyses centered on single punches, punch pairs, or trajectories; additional analyses to calculate the significance of spatial position on a cell type; and integration with the snRNA-seq and scTHS-seq data from Lake *et. al.* 2017.

Additional updates include restructuring of the manuscript to make each results section shorter and the breaking up of figures to have more, smaller figures. We believe that this restructuring focuses the manuscript and allows a reader to move between specific sections and concepts more readily.

We include a version of the manuscript with changes tracked in addition to a version with all changes accepted. Below we include a point-by-point response.

Reviewer #1 (Remarks to the Author):

The revision has substantially improved the original manuscript in: 1, quality control for the sciMAP-ATAC sequencing data, 2, punch-wise analysis results, 3, figure styles and colors, as well as reference to figures in the main text. But still some problems remain.

1, Extended figure 2. quality control. Please also show fragment length distribution. Do you observe a pattern of “nucleosome wave”? This is often used as indicative of ATAC-seq library quality.

We thank the reviewer for this suggestion and now include the insert size distributions in Extended Data Figure 2c. We observe clear nucleosome banding, with the most prominent multinucleosome banding present in the human VISp and murine ischemia datasets. The other healthy mouse brain samples produce less multinucleosome banding with a prominent subnucleosome band (reads in this band are instances where both Tn5 insertions occurred within the same accessible region without an intervening nucleosome) and mononucleosome band. We have observed over the course of hundreds of bulk and single-cell ATAC-seq experiments that the proportion within each category of banding is not indicative of quality, but that the presence or absence of banding (specifically the mononucleosome band) is the most important. Notably, the 10X Genomics scATAC technology produces a much smaller fraction of multinucleosome fragments from what we have observed, whereas a standard bulk ATAC-seq preparation produces a far greater fraction; however, both can be of high quality.

2, the analyses in figure 2j and 2k are punch-wise, which is very nice. But these are just two examples. Is it a way to invent a systematic analysis that covers all punches? More importantly, how do you know that these differences between punches are real? How to relate these findings to biology? What do these analyses predict?

We thank the reviewer for the positive comments on our punch-wise analysis. We note that we produced punch-wise data for every individual punch, both in the UMAP space (Extended Data File 2), and that the cell type composition data in Figure 2g is also punch-wise. The previous Extended Data Figure 3 built on this further, with isolated analyses on individual punches, which we now feature in Figure 3, which is focused specifically on punch-wise analysis.

We agree with the reviewer that additional analyses can be done, and performed the same spatial comparison, specifically in glutamatergic neurons, across every pair of outer and inner cortical punches. We now include a new panel, Figure 3h, where we show motif accessibility enrichment between these pairs for several transcription factor motifs.

We also note that individual punch, punch-pair, and/or trajectory analyses were performed on the human VISp and the mouse cerebral ischemia datasets. We also expanded upon these to include more detail systematically across the entire set of punches.

Finally, to confirm that these observations are indeed real; we first focused on cortical lamination which has been extensively characterized to ensure that the observations we make are consistent with the expected patterns that have been established from other methods. Our results show that we recapitulate known biology – *e.g.* enrichment of transcription factor motifs for factors known to be present in the outer or inner cortical

layers that have been established from immunofluorescence studies, such as TBR1, which we highlight – however, we reach those observations using sciMAP-ATAC, as opposed to other technologies that originally produced those observations. In our analysis of cerebral ischemic injury there are also a number of known patterns that have been identified and characterized through imaging techniques, which we also capture, lending confidence that the novel epigenetic properties that we identify are likely real and warrant further study. We note that of the top 15 spatially-significant motif enrichments in the ischemic hemisphere, 10 have previously been reported as being associated with cerebral ischemia, and 4 are associated with the ischemia associated processes of hypoxic stress, inflammation, and neurodegeneration. Our technology provided characterization of those observations for a different modality – *i.e.* chromatin accessibility versus immunofluorescence – as well as the identification of other epigenetic patterns that track with space, highlighted in Figure 8.

3, continue with the above, generally the punch-wise analyses are still limited, and often descriptive. Is it possible to some systematic analyses that really use the information from individual punches?

As we describe above, we now include additional results where we systematically analyze individual punches, pairs of punches, or trajectories across all sets. We also include a Moran's I test to calculate the significance of spatial patterning in a cell-type specific manner which is only possible using the spatial data we have produced (more details below).

4, figure 3a-c, still, the results lack of visualization and discussion in punch-level. At least, is it possible that the authors focus on the outmost punches of all trajectory and see how punches of the same layer but different positions vary?

We agree that in figures 3a-c the punch-level information is not conveyed; however, coloring the cells by individual punch is uninterpretable (188 colors). To address this, we also now highlight a panel from the updated Extended Data File 2 (described below), which represents cells from a single trajectory and all others in gray, and then a subsequent plot showing a single punch from that trajectory, with all other cells gray.

b. UMAP of cells colored by position within their respective trajectory. Top right shows the same UMAP with all cells grayed out with the exception of cells from the third trajectory from section 2. Bottom right shows all cells grayed out with the exception of cells from a single punch; the outermost cortical position (1) from the third trajectory of the second section.

Additionally, we improved on Extended Data File 2 which plotted each trajectories' punches on the UMAP with all other cells in gray; however, in the previous version all cells from the individual trajectory were the same color. We now plot them using the individual punch positional color scheme (as in the top right portion of Fig 4b above).

5, still figure 3b, it is interesting to see different punch orders mix randomly in the cluster Olig (define this!) but display a pattern in the cluster Glut. Is there any biology?

First, we thank the reviewer for pointing out that we used 'Olig' and 'Oligo' in two different figures – we now use consistent abbreviations and define the abbreviations specifically in the figure legend.

The glutamatergic neuron cluster indeed has the greatest spatial patterning. This is to be expected given cortical lamination, and suggests that other cell types within those regions exhibit far less spatial patterning. However, other cell types also exhibit a statistically significant spatial patterning; though far less than in glutamatergic neurons as quantified using a Moran's I Test (Methods; new Extended Data File 3).

We also want to highlight a new analysis where we integrate our sciMAP-ATAC data with scRNA-seq data from the human VISp (Lake *et al.* Nat. Biotech. 2018, as well as with scTHS-seq data), where our spatial positioning for glutamatergic neurons correspond to individual excitatory neuron subtypes as defined by transcriptional profiles.

Figure 5 | Integration of sciMAP-ATAC with snRNA-seq and scTHS-seq human VISp datasets. *a.* Co-embedding of sciMAP-ATAC and scTHS-seq cell profiles from Lake et. al. 2017 (ref29) using Signac⁶³ in a joint UMAP. Top right shows only scTHS-seq cells colored by cell type identified in Lake et. al. 2017, and bottom shows sciMAP-ATAC cells colored by our called cell types, with glutamatergic neurons colored by spatial position (Glut = Glutamatergic (excitatory) neurons, GABA = GABAergic (inhibitory) neurons, Astro = Astrocytes, Micro = Microglia, Olig = Oligodendrocytes, OPC = Oligodendrocyte Precursor Cells, Endo = Endothelial cells, NA = Not Applicable – no cell type provided). *b.* Co-embedding of sciMAP-ATAC and snRNA-seq transcriptional profiles from Lake et. al. 2017 using Signac. Top right shows only snRNA-seq cells. Abbreviations as in *a*, but with the addition of Per = Pericytes, and Glutamatergic (excitatory) neurons (Ex) are colored by subtype identified in Lake et. al. 2017. Bottom right shows only sciMAP-ATAC cells, with Glutamatergic neurons colored by spatial position. *c.* Confusion matrix representing the percent agreement in predicting the cell type of a cell from one dataset using the other between sciMAP-ATAC and scTHS-seq cells. *d.* As in *c*, but between sciMAP-ATAC and snRNA-seq. Spatial agreement between excitatory neuron subtypes identified in the snRNA-seq data correspond to the spatial positioning of cells within our sciMAP-ATAC dataset.

6, the paper is too long. It needs to be more concise. Consider to split figures (not to put everything into one figure, which is tiring when reading), also, similar reason, consider to split some ultra-long sections.

We thank the reviewer for this suggestion, and agree that breaking down the manuscript, both with respect to results sections and corresponding figures, makes it much more digestible. We have reduced the text length in some sections, split results into additional more-focused sections, and split our original four main figures into eight smaller, focused figures, which is possible given the format flexibility of the journal. We believe that these focused sections will allow a reader to move around the manuscript to focus on, or refer back to, specific areas of interest.

Minor:

1, line 188-190, use the same name or define acronyms in the main text and figure legends. Otherwise, who knows the symbols in the figure stand for? This happens to many panels. This seems to be a general problem.

We thank the reviewer for pointing this out. We are now consistent with the abbreviations in the figures and terminology used in the main text. We also now explicitly define the abbreviation in the figure legends.

2, Figure 3f, n=168?

n=188 punches, we now clarify in the figure and thank the reviewer for the suggestion to improve clarity.

3, Lines 291-306, this introductory paragraph is too long.

We reduced the length of this paragraph; however, some components are important background to put into context the major hits that we identify in our spatial analysis.

Reviewer #2 (Remarks to the Author):

Manuscript by Thornton et al has now been revised, and the authors have added substantial amount of new data, including from primary human visual cortex. Given this substantial change and divergence from the original flow of the manuscript, I am concerned that the manuscript has drifted quite a bit away from the original purpose.

Continuing to evaluate this manuscript as a methods paper, I am not convinced that the approach presented represents a new method. Taking punch biopsies out of the tissue followed by single cell omics hardly constitutes a new method. Incorporation of covalent barcodes has been utilized extensively by the Shendure lab, and again is not in itself particularly innovative.

We thank the reviewer for their thoughts on the flow and purpose of the manuscript. We believe that the revised manuscript, which was largely driven by the reviewer comments, does a far better job of capturing the original intent. While we understand that our method builds off of prior work (i.e. combinatorial indexing methods pioneered by our group in collaboration with the Shendure Lab and many others), bringing those innovations together along with novel workflow components to establish a working method to achieve spatial single-cell epigenetics data (which has not yet been reported on in any context) is something we believe falls into the category of a novel method.

To put our technique into a broader perspective – we achieve single-cell ATAC-seq profiles from cells derived from small tissue volumes that are spatially tracked where the resolution is ~214 microns in diameter. This is not something that has ever been published on – nor any spatially-resolved ATAC or single-cell ATAC method. We also note that early spatial RNA-seq methods achieved similar spatial resolution; however, they produced (and still produce) bulk profiles over that resolution. We achieve true single-cell profiles for multiple cells within each spatially mapped position.

We agree that we achieve this by a relatively straightforward strategy – which we argue is the greatest strength of this technique. It does not require any new equipment or other hurdles to implement, making it more accessible to the scientific community. Furthermore, this approach would not have been possible without the advancements we have put forth for isolating nuclei in a plate-based format. For any group that has deployed ATAC-seq, the assay itself is straightforward – the isolation of quality nuclei is by far the most challenging component, which is multiplied when processing a large number of samples simultaneously. The method we put forth here achieves this, and in a way that is compatible with single-cell downstream processing, which has even greater constraints than for bulk ATAC-seq assays.

We also want to thank the reviewer for their additional comments and suggestions below – particularly integration with the snRNA-seq and scTHS-seq datasets produced in Lake *et. al.* 2017, which we believe strengthen the manuscript substantially and further highlight our spatial patterning of glutamatergic neurons.

In the revised manuscript, there are now three datasets presented, mouse somatosensory cortex, human visual cortex, and mouse ischemia model. It is unclear why the authors decided to focus on these three specifically, and what unique advantages they bring. Evaluating this manuscript as a resource paper, there are several valuable resources generated that would be beneficial to the research community, but are relatively underdeveloped. For example, given that the authors now include datasets from the cerebral cortex of two species, I would expect the authors to perform some form of comparative analysis to comment about the spatial organization, or epigenomic similarity across species. This to my knowledge is the first paper that would perform such analysis.

We selected these three applications for several reasons. The first is that we wanted to apply the technique to a system that is well-studied with respect to spatial cell type composition – the cerebral cortex. The mouse

experiment was to demonstrate the feasibility of the technology and validate performance in that system, and the human dataset was to show that the methodologies we developed can be applied to postmortem human samples, and again show that we capture the progressive cell type compositions of the cortex. We also note that the spatially-mapped dataset enabled us to perform analyses of this tissue that would not have been possible – including the analysis of the spatial patterning using a Moran's I spatial autocorrelation test (Methods; new Extended Data File 3) which allowed us to calculate the significance of the “spatial progressiveness” within cell types; as well as the analysis of the spatial patterning of glutamatergic neurons integrated with the snRNA-seq from Lake *et. al.* 2017 (more below).

In the initial tissue biopsy optimization, the authors generate a dataset consisting of 8,011 cells sequenced at 12,052 reads per cell. The authors note that this compares well to the sci-ATAC-seq library, which the method is originally based on. This suggests that the method of nuclei isolation upstream of nuclei processing does not appear to negatively impact data quality. They also compare four dissociation methods, and note that the methods can vary substantially - it would be helpful to the readers to comment on how their nuclei isolation method compares to the recent paper from Slyper *et al* (Nature Medicine PMID: 32587393). Given the updated details regarding biopsy preparation, I am curious what percentage of cells/nuclei are lost in the processing - on page 4 the authors indicate that a mean of 112 cells pass quality control metrics per punch.

The reviewer is correct that the nuclei isolation process does not appear to affect downstream quality. As far as efficiency at the nuclei isolation stage, we achieve an efficiency high enough to produce more nuclei than we can physically process (*e.g.* ~10-20,000 from a single punch, which is ~1.5 million nuclei for a set of just 96 punches). For the transposition stage, we have losses of approximately 50%, which is in-line with every other existing single-cell ATAC-seq method (*e.g.* 10X Genomics). This loss results from the transposition of intact nuclei which causes a large portion to rupture. We then also incur losses during the flow sorting of tagmented nuclei into plates for the second tier of indexing via PCR (as with any flow sorting method). These losses are highly variable based on the amount of debris observed in the prep and how many sorted events it takes to properly set up gating; however, they are generally minimal. During sorting, we rarely exhaust our supply of tagmented nuclei, therefore we end up discarding substantial portions of sample. This is simply due to the fact that we prepare a specified number of PCR plates in advance to receive sorted nuclei and once we sort into all of the prepared plates, we stop. This is analogous to something like a 10X Genomics experiment, where one may have prepared millions of cells; however, each channel on the instrument that can take ~10,000 nuclei is quite expensive and one can only process so many.

Given these constraints, we cannot confidently provide a true estimate of what could be achieved with respect to efficiency; however, we are at a position where we have more tagmented nuclei than we are able to physically (or more specifically - fiscally) process.

While we believe that Slyper *et. al.* is an incredible resource for single-cell and single-nucleus isolation methods, and one we are relying on heavily for other projects in the lab, it is important to note that the protocols are very specific to sc/snRNA-seq. The buffer compositions do not generally translate well across methods, with many of the scRNA-seq (as well as scDNA-seq) isolation buffers containing salts that can disrupt chromatin and result in the ablation of ATAC signal. The isolation methods we use for ATAC are specifically designed to preserve nucleosome positioning. These buffers can be used for snRNA-seq, though we generally observe a reduction in quality when compared to the use of buffers such as those recommended by Slyper *et. al.*

There are also several concerns regarding the new analysis introduced in this paper, which is the human brain tissue. The source of the tissue is the Oregon Brain Bank, and they indicate that the PMI was 5.5 hours, and that the sample came from a 60 yr. male. Due to the lack of a biological replicate (N=1), it is impossible to assess the reproducibility of their findings. In order to overcome this clear limitation, it will be important to leverage previously published datasets, such as the one generated by Kun Zhang's lab (Lake *et al.* 2018 PMID: 29227469).

We understand that the reviewer is skeptical about the quality of our data; however, we provide extensive quality control metrics, most importantly TSS Enrichment, which measures our ability to capture the ATAC signal, which is well above the “ideal” threshold as put forth by the ENCODE Project. We also note a high FRiS and prominent nucleosomal banding pattern (all metrics presented in Extended Data Figure 2).

We agree with the reviewer that integration with publicly available data – specifically the datasets from Lake *et. al.* is an excellent idea. It is important to note that the chromatin accessibility data provided in that publication is

scTHS-seq, which is different from ATAC. Notably it requires a single Tn5 insertion event to achieve a viable library fragment, thus reducing the physical constraints on library fragment generation which produces a somewhat different signal profile.

Even with those differences, we performed integration with both the scTHS-seq dataset and the scRNA-seq dataset which are presented as a new results section and new figure (below). We note that we were able to achieve high concordance with respect to cell type agreement when predicting cell type labels across the two datasets, both for THS and RNA. This is captured in the confusion matrixes of the figure (Figure 5c,d). Of particular interest is the layer-specific excitatory neuron subtypes identified in the scRNA-seq dataset that track with the spatial information produced by our sciMAP-ATAC dataset within the glutamatergic neuron clusters.

Figure 5 | Integration of sciMAP-ATAC with snRNA-seq and scTHS-seq human VISp datasets. *a.* Co-embedding of sciMAP-ATAC and scTHS-seq cell profiles from Lake et al. 2017 (ref29) using Signac63 in a joint UMAP. Top right shows only scTHS-seq cells colored by cell type identified in Lake et al. 2017, and bottom shows sciMAP-ATAC cells colored by our called cell types, with glutamatergic neurons colored by spatial position (Glut = Glutamatergic (excitatory) neurons, GABA = GABAergic (inhibitory) neurons, Astro = Astrocytes, Micro = Microglia, Olig = Oligodendrocytes, OPC = Oligodendrocyte Precursor Cells, Endo = Endothelial cells, NA = Not Applicable – no cell type provided). *b.* Co-embedding of sciMAP-ATAC and snRNA-seq transcriptional profiles from Lake et al. 2017 using Signac. Top right shows only snRNA-seq cells. Abbreviations as in a, but with the addition of Per = Pericytes, and Glutamatergic (excitatory) neurons (Ex) are colored by subtype identified in Lake et al. 2017. Bottom right shows only sciMAP-ATAC cells, with Glutamatergic neurons colored by spatial position. *c.* Confusion matrix representing the percent agreement in predicting the cell type of a cell from one dataset using the other between sciMAP-ATAC and scTHS-seq cells. *d.* As in c, but between sciMAP-ATAC and snRNA-seq. Spatial agreement between excitatory neuron subtypes identified in the snRNA-seq data correspond to the spatial positioning of cells within our sciMAP-ATAC dataset.

The analysis of the data is remarkably superficial. The authors note that FOS motif is enriched in upper layers, NEUROD6 is enriched in lower layer, and RORB is enriched in 'middle punches'. I am not aware of any evidence that FOS expression is enriched in upper layers of the cerebral cortex, and the fact that this is an immediate early gene raises the possibility of uneven tissue degradation or some other technical artifact. As the authors mention in their description of ischemic mouse data, FOS expression may indicate neuronal damage.

The reviewer makes a good point regarding FOS motif accessibility enrichment. We selected FOS based on Gray *et. al.* (Tasic lab, <https://elifesciences.org/articles/21883>), where they identified CUX1, EGR1, FOS, MEF2C, NEUROD6, POU3F2, RFX3 and RORB motifs as being highly enriched in upper layers (defined as "L2-3 and L4"), from dissected cortical layers, from 500-cell/tube bulk ATAC-seq. However, we agree with the reviewer that FOS can be problematic. We have removed the FOS motif and now include FOXP2 as a representative motif.

Another major concern in this analysis is the profound depletion of astrocytes. Astrocytes to neuron ratio in the human visual cortex (please indicate specifically what area was profiled) should be approximately 0.6 to 1 depending on what area was profiled. The authors should evaluate the estimates of abundance of these major cell types using adjacent tissue samples by immunostaining. This should not be difficult given that they only annotate major cell classes, not cell types (please note that for clarity).

Our astrocyte to neuron ratio is 0.15:1, which is, of course, lower than 0.6:1. However, we note that our ratio is very similar (slightly higher) than the one produced by Lake *et. al.* for their assays, which produced a ratio of 0.12:1.

Taken together, the human cortex dataset featured in this revised manuscript falls short of making a compelling case for their method, and the lack of biological replicates raises the question about the quality of the data obtained from this post mortem sample.

We respectfully disagree with the reviewer, and hope that the integration with the data presented in Lake *et. al.*, as suggested, make a compelling case that the data produced are of high quality. Specifically, the cell type agreement with scRNA-seq layer-specific excitatory neuron subtypes and the spatial positioning of our sciMAP-ATAC excitatory neurons, which brings together both data quality (ability to integrate with scRNA-seq) and spatial resolution. This is coupled with high overall scores with respect to standard ATAC quality metrics (TSS Enrichment, FRiS, nucleosome banding), which are universal metrics, irrespective of sample, and well above the 'ideal' range set forth by the ENCODE Project

Reviewer #2 (Remarks to the Author)

The revised manuscript by Thornton et al represents a significant improvement over the previously evaluated version. In particular, joint analysis of data from Lake et al. reinforces the point that data quality generated with their approach is qualitatively comparable to previously published data. Despite these improvements, there are several shortcomings regarding the analysis and interpretation that would be important to address.

My concern about naming this a novel method stands unchanged. As the authors point out in their response to my critique, their paper represents an application of an existing technology to samples that are spatially recorded by an investigator. In the same vein, application of gene expression profiling using microarrays to laser captured microdissected data from the human brain generated an immensely powerful dataset featured by the Allen Institute, but was never considered to be a new method. The argument that the authors are making is completely irrational. In the same way, collecting sci-ATAC data from cells from a temporal culture sequence should not fall under the definition of a new method. It is still the same method, sci-ATAC, applied to a well-established sampling strategy, not a new method.

Secondly, even though the integration of the Lake et al. ameliorated my major concern about the lack of biological replicates, I have some concerns about the interpretation of the analysis of the integrated dataset. In particular, the authors highlight the resolution they were able to achieve with sci-ATAC with regards to cortical layers that were not seen in THS data. However, it seems that inhibitory interneurons are not resolved as well using sciATAC data as using RNAseq (from Lake et al). There are well-established distinctions in spatial distributions of MGE and CGE derived interneuron subtypes, and these populations can be easily distinguished using chromatin accessibility (see for example Mo et al. Neuron 2015 PMID: 26087164), therefore the difference is not likely due to enzymatic differences. Why is this the case? This is a relatively major issue, because interneurons are highly diverse and establish very distinct molecular signatures. If sciATAC does not recover those distinctions, this obvious shortcoming should be very clearly stated in the discussion.

Reviewer #3 (Remarks to the Author)

The revision is improved and has addressed my questions/concerns.

Reviewer #2 (Remarks to the Author):

The revised manuscript by Thornton et al represents a significant improvement over the previously evaluated version. In particular, joint analysis of data from Lake et al. reinforces the point that data quality generated with their approach is qualitatively comparable to previously published data. Despite these improvements, there are several shortcomings regarding the analysis and interpretation that would be important to address.

My concern about naming this a novel method stands unchanged. As the authors point out in their response to my critique, their paper represents an application of an existing technology to samples that are spatially recorded by an investigator. In the same vein, application of gene expression profiling using microarrays to laser captured microdissected data from the human brain generated an immensely powerful dataset featured by the Allen Institute, but was never considered to be a new method. The argument that the authors are making is completely irrational. In the same way, collecting sci-ATAC data from cells from a temporal culture sequence should not fall under the definition of a new method. It is still the same method, sci-ATAC, applied to a well-established sampling strategy, not a new method.

We respectfully disagree with the reviewer and believe that this is a philosophical disagreement with respect to what constitutes a method. We are of the view that any novel process to produce information not previously able to be obtained meets that standard. While it may not be emphasized, we conducted rigorous experiments to enable the acquisition of quality nuclei from these tissue punches – in addition to the optimization of cryosectioning at the thicknesses and variation we required without artifacts and tissue loss. We also note that many methods papers are comprised of rigorous optimizations of buffers / workflows – for example the development of the “Omni-ATAC” protocol, which ultimately arrives at a buffer formulation, but is a valuable method. We had to perform comparable buffer optimizations to enable the sciMAP-ATAC workflow.

Secondly, even though the integration of the Lake et al. ameliorated my major concern about the lack of biological replicates, I have some concerns about the interpretation of the analysis of the integrated dataset. In particular, the authors highlight the resolution they were able to achieve with sci-ATAC with regards to cortical layers that were not seen in THS data. However, it seems that inhibitory interneurons are not resolved as well using sciATAC data as using RNAseq (from Lake et al). There are well-established distinctions in spatial distributions of MGE and CGE derived interneuron subtypes, and these populations can be easily distinguished using chromatin accessibility (see for example Mo et al. Neuron 2015 PMID: 26087164), therefore the difference is not likely due to enzymatic differences. Why is this the case? This is a relatively major issue, because interneurons are highly diverse and establish very distinct molecular signatures. If sciATAC does not recover those distinctions, this obvious shortcoming should be very clearly stated in the discussion.

The reviewer is correct that we did not split out the interneuron populations in the presented figures; however, we do resolve the MGE and CGE interneuron populations as well as established interneuron sub-classifications (https://celltypes.brain-map.org/rnaseq/human_m1_10x). We note that in Figures 5a and 5b there are two distinct groupings of cells within our GABA-labeled cells, which does indeed correspond to the MGE and CGE split. We did not emphasize interneurons in the presentation of the data since it lacks the layer-specific spatial patterning that is present in excitatory neurons. We do acknowledge

that interneurons migrate from spatially distinct region of the ganglionic eminence, which we are able to identify in our data, however, research is ongoing in order to define layer biases of cortical interneurons, based on paired morphological, electrophysiological, neurochemical properties, as summarized in Lim et al. (Neuron, PMID: 30359598). We agree with the reviewer that it is important to point out that we are able to resolve interneuron subclassifications and now include a note in the main text and an additional panel in Extended Data Figure 4.

Addition to main text:

“Further subclustering of GABAergic interneurons revealed minimal spatial bias across four distinct subtypes comprised of two MGE-derived and two CGE-derived clusters (Extended Data Figure 4b-e).”

Additional Extended Data Figure panels:

New Legend Text:

“b. UMAP of the full dataset with all cells grayed out except for those identified as GABAergic neurons. c. UMAP of GABAergic neurons analyzed using topic modeling individually colored by punch position. d. Four interneuron clusters identified, including two MGE-derived and two CGE-derived cell types. e. Aggregate ATAC-seq profiles for marker genes for each of the interneuron cell types.”

Reviewer #3 (Remarks to the Author):

The revision is improved and has addressed my questions/concerns.

We thank the reviewer for their prior constructive critiques that we believe greatly improved the manuscript.